**Data Availability Statement:** All relevant data are within the manuscript and its Supporting Information files.

# Umbilical cord characteristics and their association with adverse pregnancy outcomes: A systematic review and meta-analysis

Dexter J. L. Hayes[1]*, Jane Warland[2], Mana M. Parast[3], Robert W. Bendon[4], Junichi Hasegawa[5], Julia Banks[1], Laura Clapham[1], Alexander E. P. Heazell[1]

1 Tommy's Stillbirth Research Centre, University of Manchester, Manchester, United Kingdom, 2 University of South Australia, Adelaide, Australia, 3 University of California, San Diego, CL, United States of America, 4 Retired from Norton Children's Hospital, Louisville, Kentucky, United States of America, 5 St Marianna University School of Medicine, Kawasaki, Japan

* dexter.hayes@manchester.ac.uk

## Abstract

### Objective

Current data on the role of the umbilical cord in pregnancy complications are conflicting; estimates of the proportion of stillbirths due to cord problems range from 3.4 to 26.7%. A systematic review and meta-analysis were undertaken to determine which umbilical cord abnormalities are associated with stillbirth and related adverse pregnancy outcomes.

### Methods

MEDLINE, EMBASE, CINAHL and Google Scholar were searched from 1960 to present day. Reference lists of included studies and grey literature were also searched. Cohort, cross-sectional, or case-control studies of singleton pregnancies after 20 weeks' gestation that reported the frequency of umbilical cord characteristics or cord abnormalities and their relationship to stillbirth or other adverse outcomes were included. Quality of included studies was assessed using NIH quality assessment tools. Analyses were performed in STATA.

### Results

This review included 145 studies. Nuchal cords were present in 22% of births (95% CI 19, 25); multiple loops of cord were present in 4% (95% CI 3, 5) and true knots of the cord in 1% (95% CI 0, 1) of births. There was no evidence for an association between stillbirth and any nuchal cord (OR 1.11, 95% CI 0.62, 1.98). Comparing multiple loops of nuchal cord to single loops or no loop gave an OR of 2.36 (95% CI 0.99, 5.62). We were not able to look at the effect of tight or loose nuchal loops. The likelihood of stillbirth was significantly higher with a true cord knot (OR 4.65, 95% CI 2.09, 10.37).

**Funding:** We are grateful to the STAR Legacy Foundation for providing financial support for this study. The funder had no role in study design, data collection and analyses, decision to publish, or preparation of the manuscript.

**Competing interests:** The authors have declared that no competing interests exist.

## Conclusions

True umbilical cord knots are associated with increased risk of stillbirth; the incidence of stillbirth is higher with multiple nuchal loops compared to single nuchal cords. No studies reported the combined effects of multiple umbilical cord abnormalities. Our analyses suggest specific avenues for future research.

## Introduction

Umbilical cord abnormalities (UCA) usually describe situations where fetal blood flow is reduced or interrupted due to altered structure or function of the umbilical cord. UCA are associated with adverse pregnancy outcomes including stillbirth, birth asphyxia and emergency Caesarean birth. However, estimates of the contribution of UCA to these outcomes vary; for example between 3.4% to 20% of stillbirths are reported to be caused by UCA [1]. Some of the variation may be due to the use of different classification systems for stillbirth, not all of which include UCA as a cause of death.

Of the possible reported UCA, nuchal cord, where the umbilical cord is wound at least once around the fetal neck [2], has been the subject of the most studies; its incidence increases throughout gestation, peaking at birth [3, 4]. While there are reports of nuchal cord in individual cases of stillbirth [5], data from larger studies are conflicting, with some finding significant associations [5, 6] and others reporting no effect of nuchal cord on stillbirth [7, 8]. Other UCA, including true knots and cord prolapse are rarer, but are also linked to adverse outcomes; cohort studies have demonstrated associations between true knots and perinatal death and between cord prolapse and low Apgar scores [9, 10]. In addition, an excessive or reduced number of coils of blood vessels within the cord has also been associated with various adverse outcomes [11]. UCAs can also present in combination, for example true knots may occur more in longer cords which are also more prone to entanglement [12, 13], complicating the appreciation of the significance of individual abnormalities.

Variation in published results may be due to differences in study design, mode of detection (at birth or antenatal ultrasound), definitions of abnormalities, and lack of information about characteristics such as the number of loops of nuchal cord [14], tightness of cord loops or knots [15], or duration of UCA. To address these uncertainties and to better understand the association between UCA and adverse pregnancy outcomes we undertook a systematic review and meta-analysis of observational studies to describe the normal characteristics of human umbilical cord, the incidence of UCA in singleton pregnancies, and to determine the association between UCA and adverse pregnancy outcomes. We also aimed to understand potential sources of variation between studies.

## Materials and methods

The review protocol was registered with the International Prospective Register of Systematic Reviews (PROSPERO) on the 4[th] of October 2018 (CRD420180099049). The systematic review and meta-analysis were conducted according to the PRISMA guideline [16].

### Eligibility criteria, information sources, search strategy

Cohort or cross-sectional studies that reported normal characteristics of umbilical cord or the incidence of abnormalities were included in this review. Cohort studies that reported the

incidence of UCA and their relation to adverse pregnancy outcomes or case control studies that compared pregnancies with and without UCA, or that looked at the incidence of UCA in adverse outcomes were also included. Inclusion criteria were studies of singleton pregnancies after 20 weeks of gestation, without congenital abnormalities, conducted in hospital settings (secondary or tertiary centres). Studies reporting UCA in multiple pregnancies were excluded as cord entanglement is a specific complication of monoamniotic twins. Studies of vasa praevia were not included as there is a recent systematic review [17]. All other umbilical cord abnormalities were considered for inclusion in this review.

Literature searches were conducted in MEDLINE, EMBASE, CINAHL and Google Scholar to identify relevant papers published since 1960. In addition, references from articles found, conference proceedings, and bibliographies from review articles and book chapters were examined for appropriate references. Searches were initially performed in May 2018 and updated on 1st December 2019. Example search strategies for the association between cord abnormalities and adverse pregnancy outcomes can be seen in Appendix A.

## Outcomes of interest

The primary outcome for this review was stillbirth or intrauterine fetal death (IUFD), defined as death of a baby before birth and after 20 weeks' gestation (although the definitions employed in studies were anticipated to vary according to geographical location). Secondary outcomes studied were: neonatal intensive care unit (NICU) admission, preterm birth (<37 weeks' gestation), small-for-gestational-age (birthweight <10th centile or as defined by study), low birth weight at term (<2500g), low Apgar score (<7 at 1 minute and 5 minutes) and frequency of caesarean birth. These outcomes were selected because they reflect a proposed pathway through which UCA can lead to fetal death either acutely antenatally or intrapartum, or, if the UCA were present chronically, cause fetal vascular malperfusion leading to small for gestational age infants or sufficient intrapartum compromise [18], subsequent intervention in labour (Caesarean section), low Apgar score, and/or NICU admission (Fig 1). We anticipated that a positive association would be more compelling if it was associated with a number of these related outcomes.

## Study selection and data extraction

Titles and abstracts were reviewed by two authors (from DH, JB, LC, AH) to identify relevant studies and full text papers were obtained. Data were extracted by two authors using a pre-piloted data extraction form (from DH, JW, RB, MP, JB, LC, AH); disagreements were resolved by consultation with a third author. Studies not published in English were translated where possible. When full text was not available for a study its authors were contacted, abstracts were not included if all necessary information was not present.

## Assessment of risk of bias

Quality of included studies was assessed using the NIH quality assessment tool for observational cohort and cross-sectional studies and the NIH quality assessment tool for case control studies [19]; quality of studies was judged to be good, fair, or poor. This was tailored to best suit our review question, piloted on five studies, then assessed for all included studies by two authors as described above. Studies where data on diagnostic accuracy of antenatal ultrasound could be extracted were additionally assessed using QUADAS-2 [20], which rates risk of bias and concerns regarding applicability as high, low, or unclear.

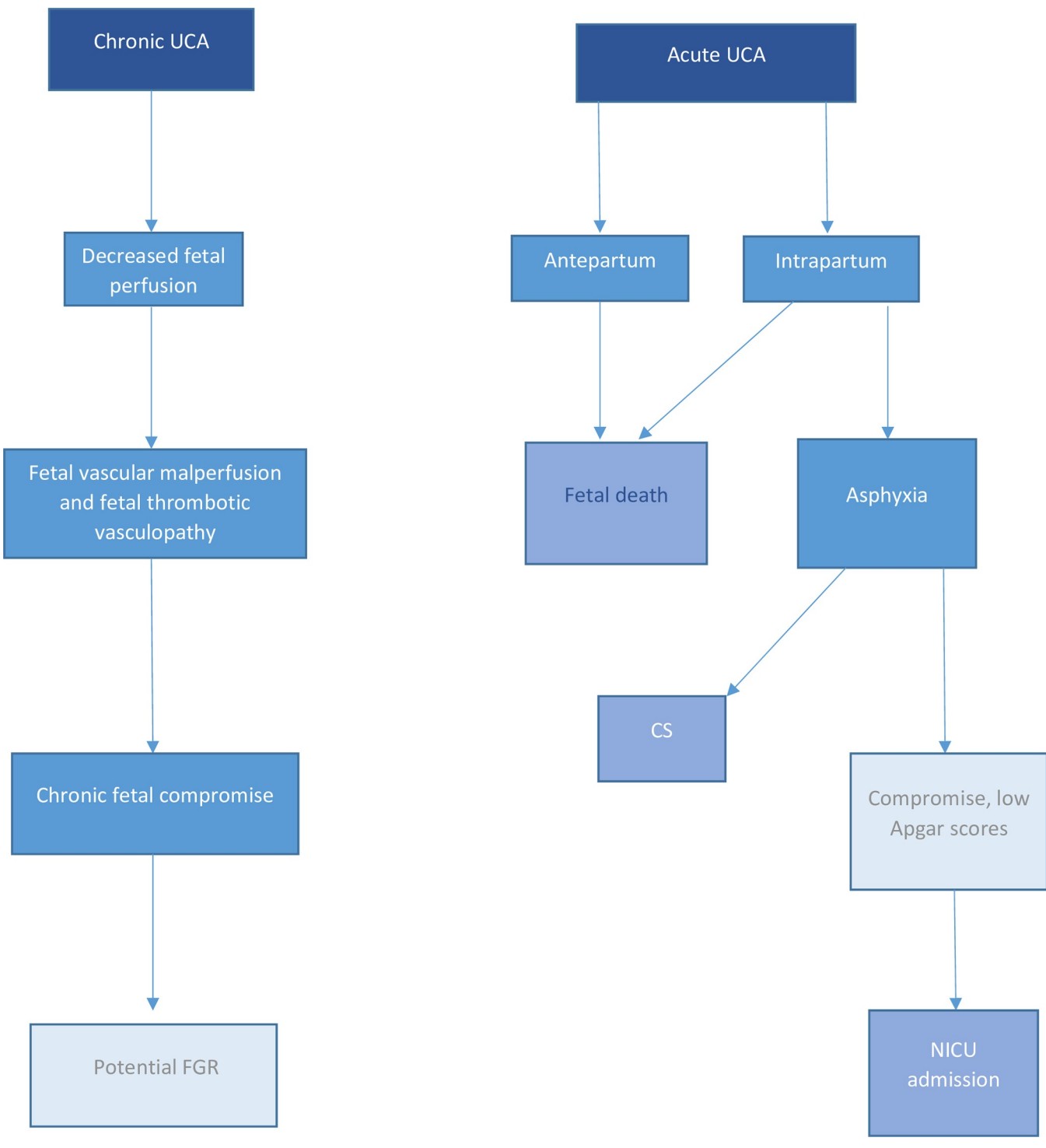

**Fig 1. Proposed pathway for potential effects of chronic and acute UCA.**

## Data synthesis

Analyses were performed using STATA version 15 [21]. Random effects meta-analysis was performed in anticipation of heterogeneity between studies due to study design. $I^2$, derived

from Cochran's chi-squared statistic Q, was calculated to describe the percentage of variability in effect estimates that is due to heterogeneity. Heterogeneity was classified as low ($I^2$ = 0–40%), moderate ($I^2$ = 41–60%), substantial ($I^2$ = 61–80%), or considerable ($I^2$ = 81–100%) [22]. Subgroup analyses were performed to investigate heterogeneity where appropriate and funnel plots were created to test for sample size effects.

Incidences of UCA were calculated using the command *metaprop* [23]. The relationship between presence of UCA and adverse outcome was investigated using the command *metan* [24]. Planned subgroup analyses were performed to examine the effects of different forms of UCA such as number of loops of cord and whether the cord could be unwound at birth. Although not originally an aim of this study, we also found papers that allowed us to calculate the diagnostic accuracy of ultrasound for detecting UCA. The STATA command *metandi* [25] was used to calculate the summary sensitivity and specificity from these studies and to produce an HSROC curve.

## Results

### Study selection and study characteristics

After screening of 2,755 abstracts, 275 full text manuscripts were assessed and 145 studies met the inclusion criteria for this review (Fig 2). Two authors [26, 27] provided further information about their studies when contacted. Key characteristics of included studies are presented in Table 1.

### Risk of bias of included studies

Quality of included studies was mostly judged to be fair: 35 studies were judged to be good and 7 poor, with the remaining 103 studies judged as fair quality using the NIH quality assessment tools. Most studies had issues with at least one of the following criteria: providing sample size justifications; measuring different levels of exposures, for example the number of loops of nuchal cord; defining exposure or outcome measures, such as the definitions of UCA or gestational age at birth or blinding of exposure and outcome assessors.

### Synthesis of results

Results are presented in four sections: normal characteristics of umbilical cord, incidence of UCA, diagnostic accuracy of ultrasound, and associations between UCA and adverse pregnancy outcomes.

**Normal characteristics of umbilical cord.** The average cord length at birth was found to be 56.0±11.1cm, using data from 39 studies of 94,849 pregnancies. Studies used a range of definitions, but if a study presented data for several different gestational age periods, then the one closest to term was used for this analysis. The average cord length at birth at 39 weeks' gestation was 55.6±12.4cm, using data from 11 studies of 13,263 pregnancies; this was chosen as it was the gestational age reported by the most studies. The mean umbilical coiling index at birth, defined as the complete number of vascular coils divided by the cord's length in centimetres [28], was 0.24±0.10 coils/cm using data from 21 studies of 8,315 pregnancies.

**Incidence of UCA.** *Nuchal cord*. The incidence of any nuchal cord at birth, determined from data from 57 studies of 830,624 pregnancies, was 22% (95% CI 19, 24). Nuchal loops combined with other entanglements were included in this analysis. Heterogeneity was considerable, $I^2$ 99.92% (p<0.001). When the number of nuchal loops were recorded (data that could be extracted only as 'multiple' and not as the exact number of loops were not included), incidences were: 1 loop 16% (95% CI 13, 19); 2 loops 3% (95% CI 2, 4) 3 loops 1% (95% CI 0, 1); 4

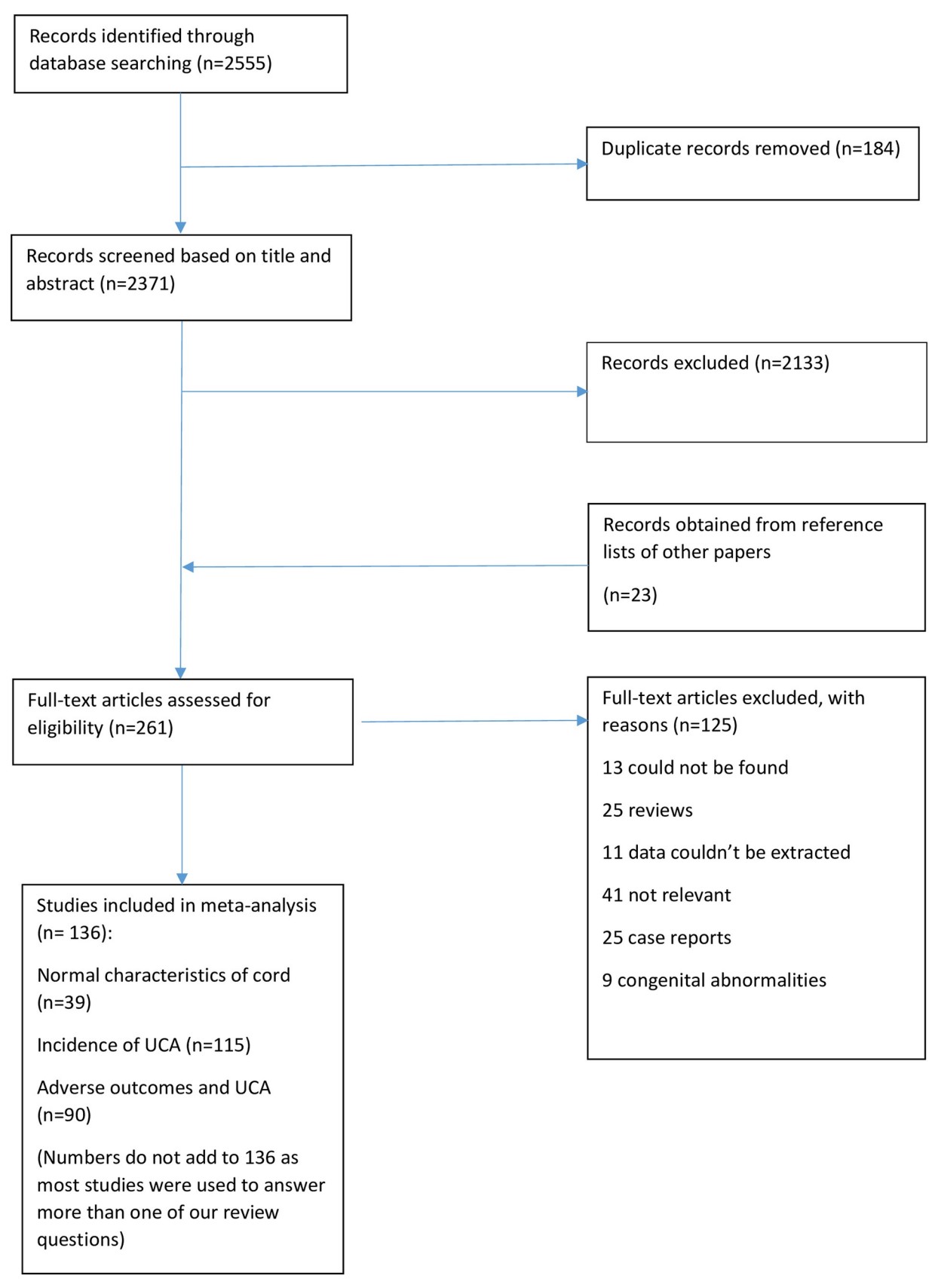

**Fig 2. PRISMA diagram.**

**Table 1. Characteristics of included studies.**

| First author | Year | Study population | Data extracted | Adverse outcomes(s) measured | Notes | Quality assessment |
|---|---|---|---|---|---|---|
| Abdallah [7] | 2018 | Prospective cohort of 455 primigravida women aged 18–35, >37w gestation. | Average cord length | CS, NICU, SB (not defined) | Sample size was reached by screening until 100 women with NC were found. | Good |
| | | | Incidence of nuchal cord | | | |
| | | Three exclusions due to entanglement of other fetal parts. | Nuchal cord and adverse outcomes | | | |
| Adesina [61] | 2014 | Cross-sectional study of singleton pregnancies, n = 428 | Average cord length | None in relation to NC, coiling | Congenital abnormalities at birth included (n = 7), significantly associated with UCI | Good |
| | | | Incidence of abnormal length | | | |
| | | | Average UCI | | | |
| | | | Incidence of nuchal cord | | | |
| | | | Incidence of cord entanglements | | | |
| | | | Nuchal cord and adverse outcomes | | | |
| | | | Abnormal coiling and adverse outcomes | | | |
| Adesina [62] | 2018 | Prospective study of 436 pregnancies | Average UCI | 5 min Apgar <7, NICU, SGA | 5 congenital anomalies included; 3 in hypercoiled group and 2 in normocoiled | Fair |
| | | | Abnormal coiling and adverse outcomes | | | |
| Adinma [6] | 1990 | Prospective cohort of 1000 consecutive births. 31 twin births included. | Incidence of nuchal cord | Apgar score, BW, fetal loss (IUFD and NND combined). | Cord entanglement was found to be more frequent at BW>2500g. Apgar score and BW presented as averages. Mortality split in to ante/intrapartum but only for all entanglements. | Fair |
| | | | Incidence of entanglements | | | |
| | | | Nuchal cord and adverse outcomes | | | |
| Agboola [63] | 1978 | Prospective study of births at 38-42w, n = 602 | Average cord length | None in relation to length | 18 IUFD and perinatal deaths in the cohort | Fair |
| Aibar [64] | 2012 | Retrospective cohort of all births at a tertiary hospital, 2003 to 2009. N = 29,530 | Incidence of true knots | None in relation to UCA | Significantly higher incidence of nuchal cord and true knots in male fetuses | Good |
| | | | Incidence of nuchal cord | | | |
| Airas [65] | 2002 | Total population at a university hospital, 1990–1999 | Incidence of true knots | 1 min Apgar <7, fetal death (not defined), LBW, NICU, PTD, SGA | TN associated with AMA, parity, previous miscarriages, obesity, male fetus, long cord | Good |
| | | | True knots and adverse outcomes | | | |
| Akkaya [30] | 2017 | Prospective case control study, 250 women with NC on US and 250 without | Nuchal cord and adverse outcomes | 1 and 5 minute Apgar score <7, BW, EmCS. | Apgar score and mode of birth did not significantly differ between groups (no. of cord entanglements). BW presented as mean. | Fair |
| Aksoy [33] | 2003 | Prospective study of 230 women who were referred to sonography for various indications, 68 of which gave birth during the study period. | Nuchal cord and adverse outcomes | None | Fetuses with severe IUGR were excluded | Fair |

(*Continued*)

**Table 1.** (Continued)

| First author | Year | Study population | Data extracted | Adverse outcomes(s) measured | Notes | Quality assessment |
|---|---|---|---|---|---|---|
| Algreisi [66] | 2016 | Retrospective cohort of term singleton births 2001–2007; n = 14,873 | Incidence of abnormal length | 1 min Apgar <7, CS, IUFD, NICU | Elective CS, preterm births excluded. Some anomalies present at birth (7%), did not differ between groups | Fair |
| | | | Incidence of cord prolapse | | | |
| | | | Abnormal length and adverse outcome | | | |
| Alnakash [67] | 2018 | Case control study, 75 women with NC at birth and 75 without. | Incidence of nuchal cord | 1 and 5 min Apgar score <7, BW, CS NICU | Incidence of single and multiple loops recorded | Fair |
| | | | Nuchal cord and adverse outcomes | | | |
| Assimakopoulos [68] | 2005 | 352 singleton pregnancies with fetuses in the vertex position. | Incidence of nuchal cord | CS | Detection of NC not used as an indication for births | Fair |
| | | | Nuchal cord and adverse outcomes | | | |
| Aviram [69] | 2015 | Retrospective cohort of all women who gave birth in a tertiary centre from 2008 to 2013 (n = 37856) | Incidence of true knots | Outcomes not presented in relation to NC | No relationship found between DFM and cord abnormalities | Good |
| | | | Incidence of nuchal cord | | | |
| Balkawade [54] | 2012 | Prospective study of 1000 women | Average cord length | Not presented in relation to NC | Preterm births excluded. Cord length significantly higher in NC. | Fair |
| | | | Incidence of cord prolapse | | | |
| | | | Incidence of abnormal length | | | |
| | | | Incidence of nuchal cord | | | |
| | | | Incidence of true knots | | | |
| | | | Abnormal length and adverse outcomes | | | |
| | | | Nuchal cord and adverse outcomes | | | |
| Balsak [70] | 2017 | Retrospective case control study of nuchal cord (477 cases, 1043 controls) | Nuchal cord and adverse outcomes | Apgar score, BW, CS, IUFD | Apgar score and BW presented as means for each group. | Fair |
| Bartling [71] | 1980 | Study of births of 115 fetuses | Incidence of nuchal cord | None | Variation in FHR seen in babies with UCA | Poor |
| | | | Nuchal cord and adverse outcomes | | | |
| Begum [72] | 2011 | Prospective cross-sectional study (n = 1646) | Average cord length | Apgar score, CS, IUGR, | IUFD excluded. 1.41:1 ratio of male to female babies with NC | Fair |
| | | | Incidence of nuchal cord | | | |
| | | | Nuchal cord and adverse outcomes | | | |
| Behbehani [73] | 2016 | Retrospective cohort of all births, 2003–2005; n = 10,040,416 | Incidence of cord prolapse | CS | IUFD excluded. Apgar score recorded as <3 at 5 min. Incidence of prolapse increased with parity. | Fair |
| | | | Cord prolapse and adverse outcome | | | |

(*Continued*)

**Table 1.** (Continued)

| First author | Year | Study population | Data extracted | Adverse outcomes(s) measured | Notes | Quality assessment |
|---|---|---|---|---|---|---|
| Bernad [74] | 2012 | Retrospective review of hospital records from 2009 to 2010 (n = 5025) | Nuchal cord and adverse outcomes | CS, IUFD | 56% of fetuses with NC were male. All IUFD were thought to be due to NC. One CS was due to umbilical cord problems, the others were for various indications. | Fair |
| Bjoro [75] | 1981 | Study of 223 pregnancies with IUGR plus 500 controls | Incidence of true knots | None | Some fetal malformations included (n = 28) | Fair |
| Blickstein [76] | 1987 | Study of pregnancies with true knots between Aug 1983 and July 1984, n = 4650 | Average cord length | 5 min Apgar <7, SB (perinatal death before birth) | Outcomes analysed compared to a control group of 108 births | Fair |
| | | | Incidence of true knots | | | |
| | | | True knots and adverse outcomes | | | |
| Bohiltea [77] | 2016 | Retrospective study of pregnancies from 2011 to 2015, n = 18500 | Incidence of true knots | 1 min Apgar <7, NICU (in true knots only). No fetal deaths. | Outcomes presented in cases with cord knots only | Fair |
| | | | True knots and adverse outcomes | | | |
| Brant [78] | 1966 | 24,084 births between 1955 and 1965 | Incidence of cord prolapse | Outcomes only given for prolapse group | 2 deaths in prolapse group due to congenital abnormalities, 15 multiple pregnancies included | Fair |
| | | | Cord prolapse and adverse outcomes | | | |
| Carey [79] | 2000 | 13,757 consecutive births from 1991 to 1996. | Incidence of nuchal cord | SB | NC not associated with antepartum or intrapartum stillbirth | Fair |
| | | | Nuchal cord and adverse outcomes | | | |
| Carey [38] | 2003 | 13,356 births from 1991 to 1996. Fetuses with BW<300g excluded | Nuchal cord and adverse outcomes | BW, IUGR | BW did not differ between single, multiple, and no NC groups | Fair |
| Carter [80] | 2018 | 8,580 women with consecutive term singleton pregnancies >37 weeks | Incidence of true knots | None | Elective CS excluded | Fair |
| | | | Incidence of nuchal cord | | | |
| | | | True knot and adverse outcomes | | | |
| Caspi [81] | 1983 | 32,365 births from 1970 to 1979 | Incidence of cord prolapse | CS, IUFD | Six twin births included. Prolapse was an indication for CS. | Fair |
| | | | Cord prolapse and adverse | | | |
| | | | outcome | | | |
| Chaurasia [82] | 1979 | 528 cords from normal full-term births | Average cord length | None | Data from aborted fetuses and multiple births not included | Fair |
| Chholak [49] | 2017 | Prospective study of 500 women with singleton pregnancies in active labour | Average UCI | 1 min and 5 min Apgar, CS, IUGR, LBW | Hypertension, diabetes, chronic renal disease excluded | Good |
| | | | Abnormal coiling and adverse outcomes | | | |
| Chitra [83] | 2012 | Prospective study of singleton pregnancies >28 weeks over two years | Average cord length | Apgar, CS, LBW, PTB | Fetal anomalies at birth included (n = 22). Elective CS excluded. | Good |
| | | | Average UCI | | | |
| | | | Abnormal coiling and adverse outcomes | | | |

(*Continued*)

**Table 1.** (Continued)

| First author | Year | Study population | Data extracted | Adverse outcomes(s) measured | Notes | Quality assessment |
|---|---|---|---|---|---|---|
| Clapp [84] | 2003 | Prospective study of nuchal cords, n = 356 | Incidence of nuchal cord | Apgar score, CS | Percentages of nuchal cord did not differ between gestational weeks. No perinatal deaths. Apgar score presented as means. | Fair |
| | | | Nuchal cord and adverse outcomes | | | |
| Collins [5] | 2000 | Prospective observational study of 1064 births in a low risk population. | Nuchal cord and adverse outcomes | SB. No NICU admissions | Six fetal abnormalities included. | Fair |
| | | | Incidence of true knots | | | |
| D'Antona [85] | 1995 | Prospective study of women in labour, n = 37 | Nuchal cord and adverse outcomes | 5 min Apgar, BW, CS (as part of operative births), NICU | BW presented as median values | Fair |
| De Laat [86] | 2006 | Prospective study of 117 pregnancies. | Average UCI | SGA | UCI measured antenatally. | Fair |
| | | | Abnormal coiling and adverse outcomes | | SUA excluded. Coiling and torsion may have been confused in some cases. | |
| Degani [87] | 1995 | 45 consecutive women with term singleton pregnancies | Average cord length | None | Coiling measured both antenatally and at birth | Fair |
| | | | Average UCI | | | |
| Degani [88] | 2001 | Singleton pregnancies with previous birth of an SGA infant | Average UCI | SGA | 39% of the cohort were SGA at birth | Fair |
| | | | Abnormal coiling and adverse outcomes | | | |
| Devaru [52] | 2012 | 100 women with singleton pregnancies at term, Jan 2007 to Aug 2008 | Abnormal coiling and adverse outcomes | NICU, IUGR, LBW, | Three neonatal deaths among NICU admissions, 1 had hypocoiling and 2 were normocoiled | Fair |
| Dhar [39] | 1995 | 3058 consecutive births. 71 twin births included. Analysis performed on 178 NC pregnancies and 356 controls. | Incidence of nuchal cord | 1 min Apgar <7, CS, IUFD, SFD | Perinatal mortality in tight NC group 6/70 (5 IUFD and 1 NND) compared to 5/356 in controls (numbers of SB/ NND not specified) | Fair |
| | | | Nuchal cord and adverse outcomes | | | |
| Dilbaz [89] | 2006 | Retrospective study of all cases of cord prolapse in 1 year out of 16,874 births. | Incidence of cord prolapse | 5 min Apgar <7, BW<2500g, NICU | Analysed as case control, 80 cases versus 800 controls. One perinatal death in cord prolapse group. | Fair |
| | | | Cord prolapse and adverse outcome | | | |
| Duman [90] | 2018 | Case control study, 60 pregnancies with cord entanglement and 60 randomly selected controls. | Cord entanglement and adverse outcomes | 5 min Apgar <7, CS | Case control study so incidence data not used | |
| El Behery [91] | 2011 | Study of 280 consecutive women in antenatal care | Average cord length | None in relation to cord length | Pregnancy complications and EFW<10th excluded | Fair |
| Enakpene [92] | 2006 | Retrospective study of 78 studies of cord prolapse | Incidence of cord prolapse | CS | Other outcomes presented for prolapse group only | Fair |
| Ercal [93] | 1996 | Prospective study of umbilical cord from 147 pregnancies | Average cord length | 1 and 5 min Apgar <7, SGA | Four fetal anomalies included, one in hypocoiled group and three in normocoiled group | Fair |
| | | | Average UCI | | | |
| | | | Abnormal coiling and adverse outcomes | | | |
| Ernst [94] | 2013 | Study of all hypercoiled cords from pathology database | Abnormal coiling and adverse outcomes | SB | Hypocoiling not studied | Fair |
| Ertuğrul [95] | 2013 | Cohort of 1784 viable singleton pregnancies born by elective caesarean section | Nuchal cord and adverse outcomes | No data | Incidence of NC increased with gestation | Fair |

(*Continued*)

**Table 1.** (Continued)

| First author | Year | Study population | Data extracted | Adverse outcomes(s) measured | Notes | Quality assessment |
|---|---|---|---|---|---|---|
| Ezimokhai [96] | 2000 | Prospective study of 1026 singleton pregnancies | Average UCI | Mean BW, CS, IUGR, PTD | 7 cords with indeterminate or incomplete turns and 47 with incomplete data were excluded. Some congenital anomalies at birth included. 20 cords had no coiling. | Fair |
| | | | Abnormal coiling and adverse outcomes | | | |
| Gabbay-Benziv [97] | 2014 | Retrospective cohort of all births Nov 2007 to Dec 2011, n = 36,889 | Incidence of cord prolapse | Presented in relation to prolapse only | Elective CS excluded. 4 twin pregnancies included | Fair |
| | | | Cord prolapse and adverse outcomes | | | |
| Gaikwad [98] | 2013 | Singleton pregnancies in labour at term | Abnormal coiling and adverse outcomes | Apgar, IUGR, LBW, NICU | IUFD prior to presentation were excluded. Average UCI presented but no SD. | Good |
| Georgiadis [99] | 2014 | Retrospective study of 47,284 singleton pregnancies | Average cord length | None in relation to length | Congenital abnormalities and IUFDs excluded | Good |
| Ghezzi [100] | 2001 | Women undergoing routine sonography, Nov 1999 to Feb 2003 | Average cord length | Overall outcomes only | Only women with sonographically lean cords were included in the study | Fair |
| Ghosh [40] | 2008 | Prospective study of post term pregnancies (>42w), n = 202 | Nuchal cord and adverse outcomes | 1 and 5 min Apgar score <7, CS, IUFD, NICU, SGA | High rate of NC likely due to gestation | Fair |
| Gibbons [101] | 2014 | Retrospective review of 409,473 live births | Incidence of cord prolapse | None | | Good |
| Gonzalez-Quintero [34] | 2004 | Retrospective study of consecutive women with nuchal cords identified via US, n = 233 | Nuchal cord and adverse outcomes | Apgar <7 at 5 min, CS, IUFD NICU, PTD, | No differences in demographics between groups | Fair |
| Gupta [102] | 2018 | Prospective observational study, n = 700 | Incidence of nuchal cord | CS, Apgar <7 at 5, 1 min | Autopsy for the perinatal death showed signs of asphyxia | Good |
| | | | Nuchal cord and adverse outcomes | | | |
| Guzikowski [103] | 2014 | Study of 2864 birth sin a one year period | Incidence of true knots | BW, CS, IUFD | Exclusion criteria not stated. | Fair |
| | | | Nuchal cord and adverse outcomes | | | |
| | | | True knots and adverse outcomes | | | |
| Hanaoka [104] | 2002 | Prospective study of 120 normal fetuses at 36–41 weeks gestation | Nuchal cord and adverse outcomes | None | | Fair |
| Hashimoto [105] | 2003 | Prospective study of women who presented for labour, IoL, or ElCS; n = 167 | Nuchal cord and adverse outcomes | 1 min Apgar score <7, BW<2500g | Preterm births excluded | Fair |
| Hehir [106] | 2017 | Retrospective cohort from 1991 to 2010 | Incidence of cord prolapse | Perinatal deaths in prolapse only | Babies >500g only; immediate expedition of birth undertaken when a cord prolapse is diagnosed | Fair |
| | | | Cord prolapse and adverse outcome | | | |
| Henry [55] | 2013 | Retrospective analysis of birth records, n = 21933 | Nuchal cord and adverse outcomes | 1 and 5 min Apgar score, NICU admission, VLBW<1500g, | | Fair |
| Hershkovitz [107] | 2001 | Consecutive singleton pregnancies, 69139 births. Study of true knots | Incidence of true knots | BW <2500g, CS, SGA | Patients with true knot had a higher incidence of nuchal cord | Good |
| | | | Incidence of nuchal cord | | | |
| | | | True knots and adverse outcomes | | | |

(*Continued*)

**Table 1.** (Continued)

| First author | Year | Study population | Data extracted | Adverse outcomes(s) measured | Notes | Quality assessment |
|---|---|---|---|---|---|---|
| Jauniaux [108] | 1995 | Retrospective study of singleton term pregnancies, n = 2650 | Nuchal cord and adverse outcomes | 1 min and 5 min Apgar <7, CS, NICU admission | Two neonatal deaths in single NC group. Case control for nuchal cord. Three perinatal deaths in multiple loop group; all women had presented with RFM. | Fair |
| Jaya [109] | 1995 | Singleton pregnancies over a 4 month period, n = 3835 | Average cord length | None in relation to cord length | | Fair |
| Jessop [110] | 2014 | Prospective study of consecutive unselected low risk patients with singleton pregnancies | Average cord length | 1 min Apgar <7, CS, NICU | 37 cords were not long enough to assess UCI | Fair |
| | | | Average UCI | | | |
| | | | Abnormal coiling and adverse outcomes | | | |
| Jo [51] | 2011 | Retrospective study of pregnancies with US at 22–28 weeks | Abnormal coiling and adverse outcomes | 1 min Apgar <7, CS, LBW, NICU, PTD | Coiling only measured antenatally | Fair |
| Joshi [111] | 2017 | Study of term singleton pregnancies. Antenatal complications such as hypertension, PE were excluded. N = 506 | Nuchal cord and adverse outcomes | Apgar <7 at 1 min, CS | Cord was also around the trunk in three cases and upper limb in another. N of 506 was number needed to get to 100 NCs | Fair |
| Kahana [112] | 2014 | Population-based study of umbilical cord prolapse, n = 12122 | Nuchal cord and adverse outcomes | None in relation to nuchal cord | Twin pregnancies included | Fair |
| | | | | | True knots were found to be associated with cord prolapse | |
| Kalem [113] | 2019 | Prospective study of singleton pregnancies between 37 and 41 weeks | Average cord length | Presented as correlations/averages only | Live births only | |
| | | | Average UCI | | | |
| Kashanian [114] | 2006 | Prospective cross-sectional study of term pregnancies, March 2003 to July 2004 | Average UCI | 5 min Apgar <7, LBW <2500g, | Exclusions: smoking, drug use, temp >37.8°c, placenta previa, abruption | Good |
| | | | Abnormal coiling and adverse outcomes | | | |
| Katsura [115] | 2018 | 106 pregnancies with MRI scan at an average of 37.4 weeks | Average cord length | CS, IUGR | | Good |
| | | | Incidence of long/ short cord | | | |
| | | | Cord length and adverse outcomes | | | |
| Kesrouani [116] | 2017 | Retrospective study of pregnancies with nuchal cord. N = 44 | Nuchal cord and adverse outcomes | CS, IUFD, IUGR, NICU admission | One twin pregnancy included, three first trimester scans | Fair |
| Kobayashi [117] | 2015 | Retrospective analysis of medical records, all women with singleton pregnancies with attempted vaginal birth >37w from Jan 2004 to Dec 2013, n = 6307 | Nuchal cord and adverse outcomes | 1 min and 5 min Apgar <7, BW, CS | Serious complications such as hypertension or diabetes excluded. Pregnancies with neck and body loops together excluded. | Fair |
| Kong [118] | 2015 | Retrospective study of all singleton births in 2010 | Nuchal cord and adverse outcomes | 5 min Apgar <7, NND, NICU admission, | One neonatal death in nuchal cord group due to trisomy 18 | Fair |
| Lal [4] | 2008 | Prospective study of 200 consecutive singleton pregnancies | Nuchal cord and adverse outcomes | None | Numbers of nuchal cords that persisted were recorded | Good |
| LaMonica [119] | 2008 | Prospective study of 166 women chosen at random | Average cord length | None | Cord length was found to not affect likelihood of vaginal birth | Fair |
| Larson [120] | 1995 | Retrospective study of singleton term pregnancies, n = 8565 | Nuchal cord and adverse outcomes | Apgar <7 at 5 min, BW, CS, IUFD (intrapartum), | Prior CS, IUFD, abnormal fetal lies excluded | Good |

(*Continued*)

**Table 1.** (Continued)

| First author | Year | Study population | Data extracted | Adverse outcomes(s) measured | Notes | Quality assessment |
|---|---|---|---|---|---|---|
| Larson [121] | 1997 | Retrospective study of singleton pregnancies at or after 20 weeks, n = 13875 | Nuchal cord and adverse outcomes | IUFD (antepartum) | IUFD prior to admission and antepartum were recorded. Frequency of NC increased with gestational age. | Good |
| Linde [9] | 2018 | Retrospective study of singleton births in Norway from 1999 to 2013, n = 856300 | Incidence of true knots | BW<10th centile, CS, NICU, IUFD (term and preterm), PTB <37w | Some fetal malformations included | Good |
| | | | Incidence of cord entanglement | | | |
| | | | True knots and adverse outcomes | | | |
| | | | Cord entanglement and adverse outcomes | | | |
| Lipitz [122] | 1993 | Retrospective case control study of nuchal cord (n = 12,241) plus prospective study of umbilical cord complications (n = 456) | Incidence of true knots | Apgar <7 at 5 min, BW <2500g, | Study comprised of a retrospective case control study and a prospective cohort study. | Good |
| | | | Nuchal cord and adverse outcomes | | Nuchal cord defined as 2+ turns around neck. | |
| | | | | | No effect of true knot on birth weight using regression model | |
| Lolis [123] | 1998 | Births from 1992 to 1996, n = 5278 | Average cord length | None | Cord length increased with parity | Good |
| Ma'ayeh [48] | 2017 | Prospective study of 72 singleton pregnancies | UCI and adverse outcomes | 1 and 5 min Apgar <7, EmCS, PTD, SGA | Coiling measured antenatally using ultrasound and postnatally as part of placental examination. Average UCI presented but without SD. | Good |
| Machin [124] | 2000 | 1. Study of 120 consecutive singleton births | Average UCI | Outcomes only in relation to the case series of pathological examinations | Cords measuring <20cm were excluded | Fair |
| | | 2. Singleton pregnancies from abnormal outcomes referred for pathologic examination (n = 1319) | Abnormal coiling and adverse outcomes | | | |
| | | | | IUFD, IUGR | | |
| Malpas [125] | 1964 | "consecutive series of normal infants born at or near term" | Average cord length | None | | Poor |
| Mariya [126] | 2018 | Retrospective study of 2957 studies from 2008 to 2012 | Average cord length | None | ElCS and EmCS for cephalopelvic disproportion, malpresentation etc. excluded. | Fair |
| | | | Incidence of true knots | | | |
| | | | Incidence of nuchal cord | | Nuchal cords and other entanglements combined for outcome data. | |
| | | | Incidence of cord entanglement | | | |
| | | | Cord entanglement and adverse outcome | | | |
| Mastrobattista [127] | 2005 | Retrospective study of all term singleton births from April 2001 to June 2002, n = 4426 | Nuchal cord and adverse outcomes | Apgar <7 at 5 min, CS, LBW, NICU admission | Live births only, noncephalic presentations excluded | Good |
| Markov [128] | 2007 | Prospective study of 86 singleton pregnancies | Nuchal cord and adverse outcomes | None | Measurements taken at 37–42 weeks | Fair |
| McLennan [129] | 1988 | Retrospective study of labour ward logbooks, n = 1115 | Incidence of true knots | 1 min Apgar score <7, IUFD, NND | 4 neonatal deaths, two due to fetal abnormalities. IUFD from 22 to 38 weeks | Fair |
| | | | Nuchal cord and adverse outcomes | | | |
| | | | True knots and adverse outcomes | | | |

(*Continued*)

**Table 1.** (Continued)

| First author | Year | Study population | Data extracted | Adverse outcomes(s) measured | Notes | Quality assessment |
|---|---|---|---|---|---|---|
| Miser [41] | 1992 | Retrospective review of births from a six month period, n = 706 | Nuchal cord and adverse outcomes | 1 and 5 min Apgar score, BW <2500g, IUGR/SGA | No significant differences in demographics between groups. No details on tightness of cords for 86/167 | Fair |
| Mittal [47] | 2015 | Prospective study of 200 randomly selected singleton pregnancies at 20-24w, Aug 2012 to July 2013 | Average UCI | 5 min Apgar <7, CS, NICU admission, PTD | SUA excluded | Good |
| | | | Abnormal coiling and adverse outcomes | | | |
| Naeye [130] | 1985 | 35,779 singleton pregnancies from 13 centres | Average cord length | Data not presented in a way that could be analysed | Data presented for various gestations | Fair |
| Najafi [131] | 2018 | Prospective study of 296 consecutive pregnancies, Oct 2014 to August 2016 | Average UCI | None | UCI measured antenatally at 37-41w | Good |
| Narang [12] | 2014 | Cross-sectional study, n = 150 | Average cord length | Apgar <7 at 5 min, NICU admission | Single and multiple NC groups combined for analysis. | Fair |
| | | | Nuchal cord and adverse outcomes | | NC was more common in multiparous women and cord length was significantly longer in NC. | |
| | | | | | One neonatal death in an infant with one tight loop | |
| Ndolo [132] | 2017 | Prospective study of singleton pregnancies | Abnormal coiling and adverse outcomes | CS, SGA, PTD | Gestation between 18 and 24 weeks | Good |
| Nkwabong [13] | 2018 | Case control study of singleton pregnancies, n = 2015 | Nuchal cord and adverse outcomes | BW <2500g, CS, PTD | Cord length significantly higher in nuchal cords | Fair |
| Nnatu [133] | 1960 | 661 consecutive singleton pregnancies at term | Average cord length | None | No significant correlation found between parity and cord length | Fair |
| Ogueh [134] | 2006 | Retrospective study of singleton pregnancies with BW>2500g, n = 57853 | Nuchal cord and adverse outcomes | CS | Pregnancies with NC less likely to be born by CS | Fair |
| Ohno [135] | 2016 | 200 consecutive singleton term births | Average cord length | None | UCI also calculated but outcome data presented as averages | Fair |
| Olaya-C [136] | 2018 | Retrospective observational study, 2013–2014; n = 434 | Incidence of true knots | 1 neonatal death and 2 stillbirths, not associated with TN | 22 twin pregnancies excluded from final analysis | Fair |
| | | | Abnormal coiling and adverse outcomes | | | |
| Önderoğlu [8] | 2008 | Retrospective study of all births with nuchal cord from 2002 to 2004 | Nuchal cord and adverse outcomes | SB (not defined), Apgar <7 at 1 min | Term pregnancies only. | Good |
| Osak [137] | 1997 | Retrospective study of hospital records over a three year period, n = 10509 | Nuchal cord and adverse outcomes | Mean BW | Only live births were included in the study. Hypertension, IUGR, diabetes also excluded | Fair |
| Pathak [138] | 2010 | 861 women with consecutive singleton pregnancies that birthed at 37-42w | Average cord length | None in relation to cord length | | Fair |
| Patil [139] | 2013 | Prospective study of 200 patients in active labour at term | Abnormal coiling and adverse outcomes | 1 and 5 min Apgar <7, IUGR, NICU | EmCS were not included | Good |
| Peng [140] | 2006 | Retrospective review of 268 fetal autopsies | Abnormal coiling and adverse outcomes | IUFD | Hypocoiling not recorded | Fair |

(*Continued*)

**Table 1.** (Continued)

| First author | Year | Study population | Data extracted | Adverse outcomes(s) measured | Notes | Quality assessment |
|---|---|---|---|---|---|---|
| Peregrine [31] | 2005 | Prospective study of women undergoing induction after 36 weeks gestation, n = 237 | Nuchal cord and adverse outcomes | 1 and 5 min Apgar score, CS, NICU admission | No association between reduced fetal movements and NC. One neonatal death due to a congenital malformation. | Fair |
| Poljak [141] | 1989 | Study of women who had antenatal US at term before induction, n = 100 | Nuchal cord and adverse outcomes | Apgar score (mean only) | Mean Apgar was significantly lower in NC group | Fair |
| Purola [142] | 1968 | Series of 1980 consecutive singleton births with BW>600g | Average cord length | Apgar score (time not stated), LBW | No. of loops of nuchal cord recorded but not in relation to outcomes. Cord around neck in 3/25 IUFDs but not attributed as a cause–two were due to fetal abnormalities. | Good |
| | | | Nuchal cord and adverse outcomes | | | |
| Qin [32] | 2000 | Prospective study of 180 consecutive singleton pregnancies, n = 531 | Average cord length | None | Color Doppler results included in analysis, greyscale also used | Good |
| | | | Average UCI | | | |
| | | | Nuchal cord and adverse outcomes | | | |
| | | | Abnormal coiling and adverse outcomes | | | |
| Räisänen [143] | 2013 | Retrospective study of singleton births from 2000 to 2012, n = 27537 | Incidence of true knots | 1 min and 5 min Apgar <7, CS, IUFD, LBW, NICU, PTD, SGA | | Good |
| | | | True knots and adverse outcomes | | | |
| Rana [144] | 1995 | Prospective study of placentas from consecutive high risk patients | Average cord length | 5 min Apgar <7 | Fetal congenital anomalies diagnosed by day 3 of life were included | Fair |
| | | | Average UCI | | | |
| | | | Abnormal coiling and adverse outcomes | | | |
| Rayburn [145] | 1981 | 536 term births | Incidence of cord prolapse | None | UCA were most frequent with long cords | Fair |
| | | | Incidence of abnormal length | | | |
| | | | Incidence of true knots | | | |
| Rogers [146] | 2003 | Case control study for nuchal cord, n = 66 for each group | Nuchal cord and adverse outcomes | Apgar score, CS | Mean Apgar scores were significantly higher in the group with no entanglement | Fair |
| Romero Gutierrez [147] | 2000 | Prospective cross sectional study, n = 132 | Nuchal cord and adverse outcomes | 1 and 5 min Apgar score <7, BW, IUFD | Low risk pregnancies only | Fair |
| Sahoo [148] | 2015 | 177 women with singleton pregnancies and US examination at 18 to 23 weeks | Average UCI | CSFD, IUFD, PTD | High risk pregnancies (diabetes, hypertension) were excluded | Good |
| | | | Abnormal coiling and adverse outcomes | | | |
| Salge [149] | 2018 | Cross-sectional study from 2012 to 2015, n = 265 | Incidence of true knots | None | 126/265 were high risk pregnancies | Fair |
| Schaffer [150] | 2005 | Retrospective study of women with planned vaginal births, n = 9574 | Nuchal cord and adverse outcomes | 1 and 5 min Apgar score, CS, NICU admission | Mean BW significantly lower in nuchal cord groups | Fair |
| Sharma [151] | 2012 | Study of all booked singleton primigravidas in the second trimester of pregnancy | Average UCI | 5 min Apgar <7, CS, FGR, LBW, PTD | | Fair |
| | | | Abnormal coiling and adverse outcomes | | | |

*(Continued)*

**Table 1.** (Continued)

| First author | Year | Study population | Data extracted | Adverse outcomes(s) measured | Notes | Quality assessment |
|---|---|---|---|---|---|---|
| Sheiner [152] | 2006 | Retrospective study of all births from 1988 to 2003, n = 16631 | Nuchal cord and adverse outcomes | 1 and 5 min Apgar score <7, BW, CS | Perinatal mortality was significantly lower in pregnancies with nuchal cord. 1 min Apgar <7 more common in NC but 5 min less common. | Good |
| Shiva Kumar [153] | 2017 | Prospective study of 1000 term pregnancies picked at random | Average cord length | None | Pregnancies were excluded if FHR measured during labour. Knots more common in long cords. Only one nuchal cord occurred with short cord; 70 were long cords (>95cm) | Fair |
| | | | Incidence of true knots | | | |
| | | | Nuchal cord and adverse outcomes | | | |
| Shrestha [154] | 2007 | Prospective cross-sectional study of women who gave birth after 28 weeks, n = 512 | Nuchal cord and adverse outcomes | 1 and 5 min Apgar score <7, CS, NICU admission | | Fair |
| Singh [15] | 2008 | Review of labour records over a six month period, n = 350 | Nuchal cord and adverse outcomes | 1 and 5 min Apgar score <7, CS | | Fair |
| Sinnathuray [155] | 1965 | All births in 1961, n = 3917 | Nuchal cord and adverse outcomes | None | Study recorded 26 perinatal deaths, none of which were attributed to nuchal cord. 2 were due to fetal abnormalities. | Fair |
| Sørnes [156] | 1989 | 5675 births between 1979 and 1984 | Average cord length | CS | BW <3000g, ElCS, operative births excluded | Fair |
| Sørnes [157] | 1995 | 11,201 singleton births from 1991 to 1994 | Incidence of entanglements | None | 836 insufficiently filled charts excluded. Entanglements described as encirclements in the paper, not specified. | Poor |
| Sørnes [158] | 2000 | Study using obstetric database between 1991 and 1997, n = 22012 | Incidence of true knots | CS, IUFD (antepartum, death during or after birth classified as perinatal loss) | Fetal deaths before 24 weeks were not included. | Fair |
| | | | True knots and adverse outcomes | | Number of knots was recorded | |
| Stanek [159] | 2016 | Consecutive pregnancies >21 weeks from 1994 to 2013, n = 5634 | Incidence of true knots | None | Loose and tight knots recorded | Fair |
| | | | Abnormal coiling | | Some multiple pregnancies and congenital malformations included | |
| Stefos [160] | 2003 | 534 consecutive singleton pregnancies | Average cord length | None | Cord length increased with parity | Fair |
| Strong [161] | 1996 | Prospective study of 200 consecutive pregnancies | Nuchal cord and adverse outcomes | None | Mean UCI was significantly higher in nuchal cords. Characteristics of NC not recorded. | Poor |
| Suzuki [162] | 2011 | All singleton pregnancies at 34–41 weeks between 2002 and 2009, n = 10453 | Average cord length | None | True knots were more common in long cords >68cm. Chromosomal aberrations present in 7 fetuses. | Fair |
| | | | Incidence of true knots | | | |
| Tamrakar [163] | 2013 | Case control study, 289 cases with at least one nuchal loop and 965 randomly selected controls from 4219 unaffected singleton pregnancies | Incidence of nuchal cord | CS | 73% controls were birthed by CS; this paper was excluded from NC and CS analysis | Fair |
| | | | Nuchal cord and adverse outcomes | | | |
| Tantbirojn [164] | 2009 | Retrospective study of pathology database, n = 224 | Incidence of true knots | Apgar score, IUGR (not defined), IUFD (not defined) | 49% of long cords had true knots. Case control for nuchal cord so incidence not used. | Fair |
| | | | Nuchal cord and adverse outcomes | | | |
| Tapasvi [165] | 2017 | 100 singleton term births | Average cord length | 1 min and 5 min Apgar <7 | Preterm births, previously diagnosed IUFD, instrumental births excluded | Fair |

(*Continued*)

**Table 1.** (Continued)

| First author | Year | Study population | Data extracted | Adverse outcomes(s) measured | Notes | Quality assessment |
|---|---|---|---|---|---|---|
| Tripathy [166] | 2014 | Prospective study of high risk singleton pregnancies, n = 100 | Average cord length | 5 min Apgar <7, LBW, PTD | | Fair |
| | | | Average UCI | | | |
| | | | Abnormal coiling and adverse outcomes | | | |
| Uygur [167] | 2002 | 32,457 births that occurred during the study period | Incidence of cord prolapse | BW <2500g | 1.73% incidence of twin pregnancy in controls, none in prolapse group. Cord prolapse associated with multiparity | Fair |
| Van Dijk [168] | 2002 | Uncomplicated singleton pregnancies, Jan-April 2000; n = 122 | Average cord length | None | Preeclampsia, hypertension, diabetes, LBW, birth for fetal distress excluded | Fair |
| | | | Average cord length | | | |
| | | | Average UCI | | | |
| Vasa [45] | 2018 | Retrospective study of all births in 2012 at Mercy Hospital | Incidence of nuchal cord | 1 and 5 min Apgar <7, CS, IUGR, NICU | | Fair |
| | | | Nuchal cord and adverse outcomes | | | |
| Vintzileos [43] | 1992 | Retrospective study of referred high risk women over a two year period, n = 520 | Nuchal cord and adverse outcomes | 1 and 5 min Apgar score <7, CS, IUGR, PTD | Tertiary referral centre– 90% high risk patients. 379 were preterm. Perinatal death also recorded, no difference between groups. | Fair |
| Walker [169] | 1960 | Retrospective study of 223 consecutive births | Average cord length | None | No link found between cord length and parity | Poor |
| Walla [170] | 2018 | Study of 486 pregnant women, Feb 2014 to May 2016 | Incidence of true knot | CS, NICU admission | Two twin pregnancies included, neither had UCA. No stillbirths in the study. Incidence of other entanglements with nuchal cord also presented. | Fair |
| | | | Incidence of nuchal cord | | | |
| | | | Nuchal cord and adverse outcomes | | | |
| Wang [44] | 2016 | Retrospective study of medical records, n = 1749 | Nuchal cord and adverse outcomes | 1 and 5 min Apgar <7, EmCS, SGA | Terminations <22w, IUFD <37w, ElCS, PTD all excluded. NC was not routinely evaluated using sonography and did not affect management. | Good |
| Wasswa [171] | 2014 | Retrospective review of all births with cord prolapse >28w, 2000 to 2009; n = 438 | Cord prolapse and adverse outcomes | Outcomes in prolapse only: 5 min Apgar <7, CS, IUFD | 438 randomly sampled from 661 after exclusion of 273 IUFD | Fair |
| Weiner [172] | 2015 | Retrospective study of women who underwent EmCS for FHR at 37–42 weeks, n = 530 | Incidence of true knots | 5 min Apgar score <7, BW (continuous). | Significantly higher CAPO incidence in multiple loops compared to single (this includes limb and trunk entanglements). No difference in BW between groups. Case control for EmCS so not used as an outcome. | Good |
| | | | Incidence of cord entanglement | | | |
| | | | Nuchal cord and adverse | | | |
| | | | outcomes | | | |
| | | | Cord entanglement and adverse outcome | | | |
| Winch [173] | 1961 | 48,885 births over a ten year period, exclusions not stated. | Incidence of cord prolapse | Perinatal death, not defined | States that incidence may be low as high standards of documentation were not adapted until 1957 and BW<1500g was not included. | Poor |
| Wu [174] | 1996 | Prospective study of 1087 births >28w, May to Aug 1995 | Average cord length | | Cord length and entanglements had no effect on fetal distress | Fair |

(*Continued*)

**Table 1.** (Continued)

| First author | Year | Study population | Data extracted | Adverse outcomes(s) measured | Notes | Quality assessment |
|---|---|---|---|---|---|---|
| Yadav [175] | 2013 | Case control study for cord length, n = 200 | Incidence of true knots | CS | IUFDs, diabetes, PE, PTD excluded. Data could only be extracted for long cords (n = 80) | Fair |
| | | | Cord entanglement and adverse outcome | | | |
| Yamamoto [26] | 2016 | Retrospective study of singleton births | Incidence of abnormal length | 5 min Apgar <7, SGA | Average cord length was 56.6cm but SD not reported | Fair |
| | | | Abnormal length and adverse outcomes | | | |
| Zahoor [176] | 2013 | Retrospective study of labour records from 2011, n = 1776 | Nuchal cord and adverse outcomes | Apgar score, NICU admission | 85 women had ElCS due to cord around neck at term. | Poor |

**Key**: BW, birthweight; CS, caesarean section; ElCS, elective caesarean section; EmCS, emergency caesarean section; FGR, fetal growth restriction; IUFD, intrauterine fetal death; LBW, low birthweight; NICU, neonatal unit admission; NND, neonatal death; PTB, preterm birth; SFD, small for dates; SGA, small for gestational age; SB, stillbirth.

or 5 loops <1%. 32 studies of 89,455 pregnancies presented data for at least a single loop of cord. Loose nuchal loops were more frequent than tight loops, with a summary frequency of 10% (95% CI 4, 18) compared to 5% (95% CI 4, 7; data from 230,729 pregnancies from 10 studies).

The incidence of nuchal cord detected by ultrasound scan at any gestational age was 28% (95% CI 21, 36; $I^2$ 97.67%; data from 13 studies of 4,107 pregnancies). Case control studies were not included in this analysis.

It was not possible to calculate the incidence of other entanglements due to variation in study definitions and outcomes.

*Cord prolapse*. Incidence of cord prolapse was calculated from 21 studies of 11,057,165 pregnancies; the overall incidence was 0.17%.

*True knots*. Overall, the incidence of true knots at birth was 1% (0, 1). Heterogeneity was considerable($I^2$ 98.52%, p<0.001); data from 27 studies of 1,289,679 births. Only one paper [29] recorded the incidence of multiple knots; 14 were found from 22,012 births (0.06%).

*Abnormal coiling*. Twenty-one studies reported the frequency of abnormal coiling, but the incidences of hypercoiling and hypocoiling were not calculated as they were generally defined using the 90th and 10th centiles respectively.

It is important to note, however, that the actual measurements used to define these centiles differed between studies depending on the populations.

*Abnormal length*. The incidence of abnormal cord length could not be recorded due to wide variation in study definitions. Definitions of a long cord ranged from >59.0cm to >95.0cm, and definitions of a short cord ranged from <35.0cm to <50.0cm.

**Diagnostic accuracy of ultrasound.** *Nuchal cord*. We identified 12 papers which reported the diagnostic accuracy of ultrasound scanning for predicting nuchal cord at birth, these are described in the characteristics of included studies table (Table 1). For this element, a positive index test result was any nuchal cord suspected antenatally using ultrasound and the reference standard was the presence of a nuchal cord at birth. Results were combined for ultrasound screening at any gestation; four studies [7, 30–32] performed screening immediately prior to induction or during labour and in all but two studies all measurements were performed after 36 weeks [33, 34].

QUADAS-2 was used to quantify the risk of bias and applicability concerns for each included study. Most papers were at low or unclear risk of bias for all domains. Akkaya et al.

[30] was judged to be at high risk of bias for patient selection and index test domains, while Gonzalez-Quintero et al. [34] was judged to be at high risk of bias for patient selection; these were both case control studies. Studies were all low or unclear risk of bias for applicability concerns. Only one study [32] was judged to be low risk for all domains. Six studies blinded reference standard results[3, 4, 32, 35–37] all but one of these [36] also blinded index test results. All other studies did not state whether blinding took place.

Summary sensitivity for ultrasound at all gestations was 80.5 (95% CI 66.3, 89.6), summary specificity 86.6 (95% CI 80.0, 91.2). However, there was considerable variation in sensitivity of individual studies ranging from 29.0 to 96.8%, with specificities ranging from 57.0% to 96.6%. The positive likelihood ratio (LR+) was 6.01 and the negative likelihood ratio (LR-) was 0.17. Sensitivities and specificities from each study were used to produce an HSROC plot (Fig 3); the diagnostic odds ratio (DOR) for ultrasound scanning at all gestations was 26.6 (95% CI 9.46, 74.7). There did not appear to be a linear relationship between accuracy and gestational age although for studies where ultrasound scanning was performed in early labour the sensitivity values were higher, ranging from 90.2 to 96.8%.

*True knots.* The accuracy of ultrasound for the detection of true knots at birth could not be analysed due to a lack of available study data.

**Associations between UCA and adverse pregnancy outcomes.** *Nuchal cord and stillbirth.* When data for any nuchal cord at birth were pooled, no statistically significant association was detected between presence of any nuchal cord and stillbirth (OR 1.11; 95% CI 0.62, 1.98). Heterogeneity was moderate ($I^2$ 44.4%, p = 0.055). As no association was detected for a single loop of nuchal cord versus controls (OR 0.87; 95% CI 0.56, 1.35), data were combined for an analysis comparing multiple loops to combined data for no loop and a single loop. This resulted in an OR of 2.36 (95% CI 0.99, 5.62; p = 0.053) for multiple loops of nuchal cord (Fig 4). Heterogeneity for this analysis was low ($I^2$ 7.0%, p = 0.372). Comparing multiple loops to no loops, excluding single loops from the analysis, resulted in an OR of 1.91 (95% CI 0.90, 4.06). Heterogeneity was again low ($I^2$ 0.0, p = 0.623). 123 stillbirths from 40,114 pregnancies were included in the pooled analysis. There was no evidence of small study effects (Harbord's test, p = 0.137).

Three studies presented data for the relationship between nuchal cord detected by ultrasound at any gestation and stillbirth, no statistically significant association was detected (OR 0.72; 95% CI 0.17, 3.05). Heterogeneity for this analysis was low ($I^2$ = 25.8%, p = 0.260). Data from this analysis were obtained from 1955 pregnancies, 24 of which were stillbirths.

*Nuchal cord and other adverse outcomes.* Results from analyses of the relationships between nuchal cord and all secondary adverse outcomes are shown in Table 2. Analyses could not be performed for the association between nuchal cord detected using ultrasound and Apgar scores <7 at 5 minutes, NICU admission, small-for-gestational age, or preterm birth, or for the association between nuchal cord at birth and preterm birth.

A single loop of nuchal cord at birth was only associated with a 1 minute Apgar score <7 whereas multiple loops of cord were associated with increased likelihood of caesarean section and Apgar scores <7 at both one and five minutes. Tight loops of nuchal cord but not loose loops were associated with low Apgar scores.

We found no evidence for an association between nuchal cords at birth and NICU admission. Overall heterogeneity was substantial at 66.8% (p<0.01); this was due to considerable heterogeneity in the data from multiple loops ($I^2$ 88.5%, p = 0.00), whereas heterogeneity in the data for single loops was low ($I^2$ 30.5%, p = 0.22). However, the likelihood of NICU admission with a tight nuchal cord at birth was twice as high as with no nuchal cord, although this was not statistically significant.

No included studies specified nuchal cord as an indication for birth; if a study presented emergency caesarean section or caesarean section for fetal distress separately, then these data

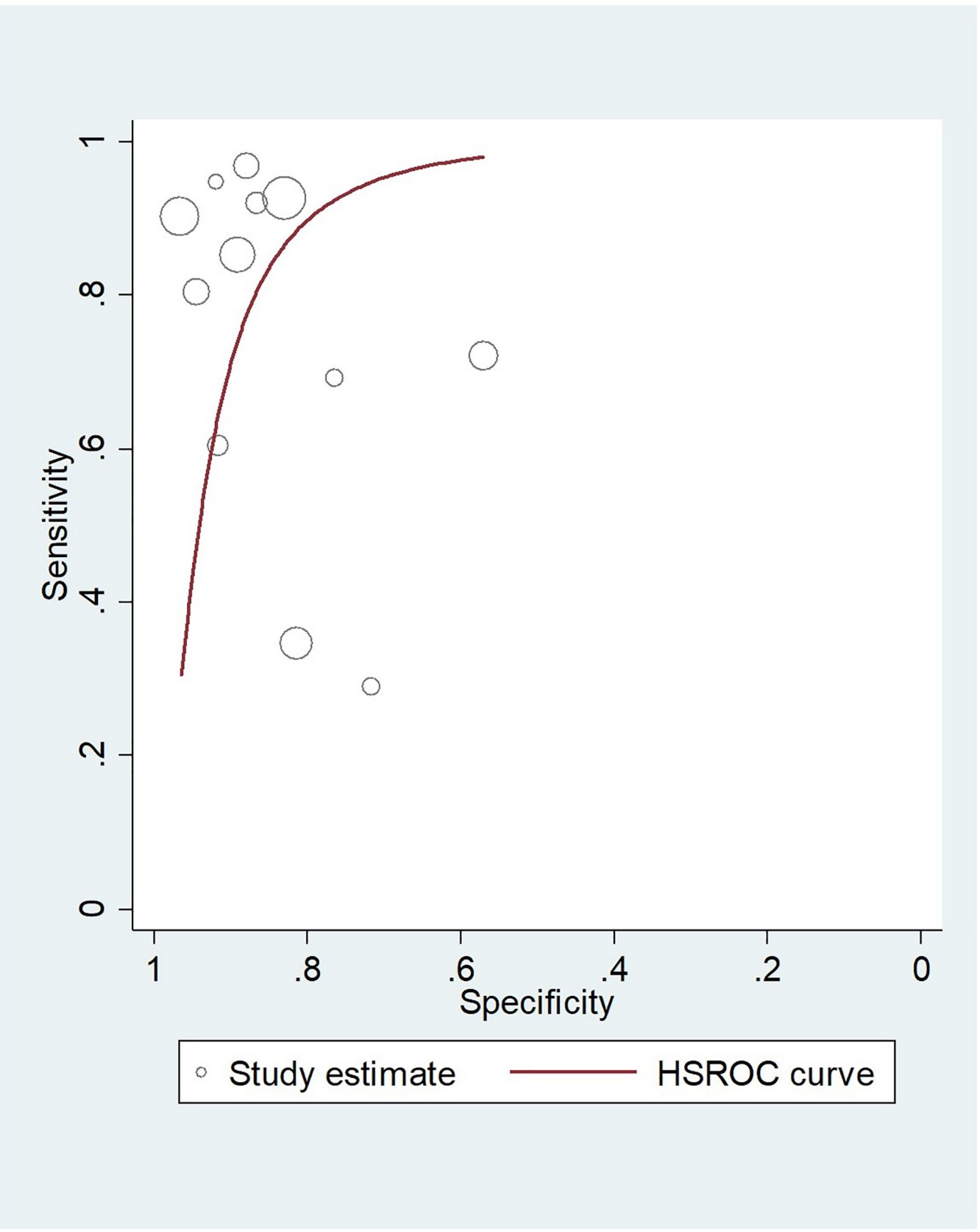

**Fig 3. HSROC plot for the diagnostic accuracy of antenatal ultrasound to predict nuchal cord at birth.**

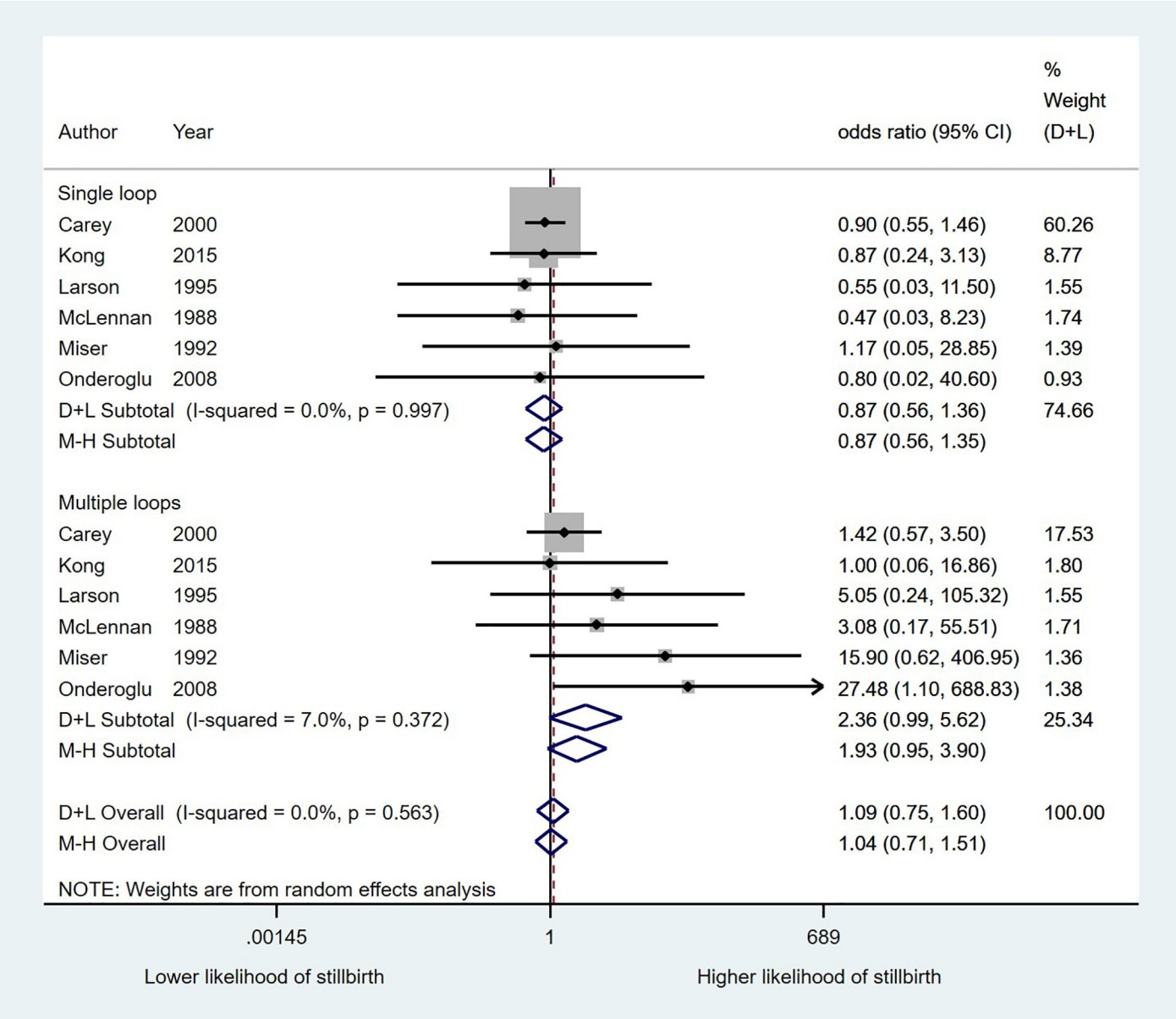

**Fig 4. The association between nuchal cord loops and the likelihood of stillbirth.**

were used instead of the overall rate. However, caesarean section was significantly more likely in pregnancies with nuchal cord detected via ultrasound (OR 1.64; 95% CI 1.07, 2.51) Heterogeneity was low ($I^2$ 33.5%, p = 0.211).

No significant relationship between nuchal cord and birth weight <2500g (OR 0.66; 95% CI 0.50, 1.35), or fetal growth restriction or small for gestational age infants (OR 1.41; 95% CI 0.90, 2.21) was identified. Studies of fetal growth restriction and small for gestational age were combined as they used a wide range of definitions [8, 38–45].

*Sensitivity analyses for nuchal cord papers.* No studies that were rated poor by quality assessment presented data for nuchal cord and its relationship to adverse outcome so planned sensitivity analyses were not performed.

**Table 2. Analyses of the relationship between nuchal cord and adverse outcomes.**

|  | Stillbirth | CS | 1 min Apgar score <7 | 5 min Apgar score <7 | NICU admission |
|---|---|---|---|---|---|
| **All nuchal cords** | | | | | |
| Odds ratio (95% CI) | 1.11 (0.62, 1.98) | 1.05 (0.88, 1.25) | **1.70 (1.31, 2.20)** | 1.12 (0.86, 1.47) | 1.17 (0.99, 1.39) |
| $I^2$ | 44.4% (p = 0.055) | 93.7% (p = 0.00) | 87.4% (p = 0.00) | 46.6% (p<0.05) | 66.8% (p<0.01) |
| Number of pregnancies | 40,114 | 274,107 | 210,102 | 210,102 | 243,712 |
| **Single loop** | | | | | |
| Odds ratio (95% CI) | 0.87 (0.56, 1.35) | **0.66 (0.50, 0.88)** | **1.80 (1.22, 2.65)** | 0.86 (0.42, 1.75) | 1.01 (0.86, 1.18) |
| $I^2$ | 0.00% (p = 0.997) | 83.8% (p = 0.00) | 54.3% (p = 0.087) | 68.7% (p<0.01) | 30.5% (p = 0.22) |
| Number of pregnancies | 28,687 | 31,230 | 17,568 | 29,718 | 21,097 |
| **Multiple loop (no NC as controls)** | | | | | |
| Odds ratio (95% CI) | 1.91 (0.90, 4.06) | **1.60 (1.10, 2.32)** | **3.39 (2.30, 5.01)** | **2.74 (1.12, 6.73)** | 1.75 (0.92, 3.34) |
| $I^2$ | 0.0% (p = 0.623) | 79.0% (p = 0.00) | 53.0% (p = 0.075) | 72.0% (p<0.01) | 88.5% (p = 0.00) |
| Number of pregnancies | 22,649 | 25,028 | 14,100 | 26,638 | 16,824 |
| **Multiple loop (single loop plus no NC as controls)** | | | | | |
| Odds ratio (95% CI) | 2.36 (0.99, 5.62) | **1.66 (1.21, 2.28)** | **2.77 (1.53, 5.03)** | 2.20 (0.75, 6.48) | 1.79 (0.92, 3.49) |
| $I^2$ | 7.0% (p = 0.372) | 71.4% (p<0.01) | 80.8% (p<0.01) | 85.8% (p = 0.00) | 88.3% (p = 0.00) |
| Number of pregnancies | 29,629 | 32,851 | 17,906 | 34,764 | 22,332 |
| **Tight loop (no NC as controls)** | | | | | |
| Odds ratio (95% CI) | Insufficient data | 1.42 (0.46, 4.41) | **6.94 (2.42, 19.59)** | **7.57 (1.80, 11.60)** | 2.27 (0.73, 7.05) |
| $I^2$ | n/a | 95.9% (p = 0.00) | 65.1% (p<0.05) | 0.0% (p = 0.657) | 85.8% (p = 0.00) |
| Number of pregnancies | n/a | 63,698 | 3468 | 2968 | 174,639 |
| **Loose loop (no NC as controls)** | | | | | |
| Odds ratio (95% CI) | Insufficient data | **0.72 (0.54, 0.97)** | 0.92 (0.61, 1.37) | **0.37 (0.14, 0.96)** | 0.86 (0.65, 1.14) |
| $I^2$ | n/a | 26.4% (p = 0.227) | 29.6% (p = 0.235) | 0.0% (p = 0.930) | 26.9% (p = 0.251) |
| Number of pregnancies | n/a | 6515 | 4131 | 3593 | 208,116 |

Shading and bold text indicate statistical significance. **Key:** CS = caesarean section. NC = nuchal cords, 95% CI = 95% confidence interval.

*True knots and stillbirth.* The likelihood of stillbirth was significantly higher in pregnancies with a true knot in the umbilical cord at birth than in those without, with an OR of 3.96 (95% CI 1.85, 8.47; 7 studies of 930,314 births) (Fig 5) Heterogeneity was moderate ($I^2$ 60%, p<0.05).

*True knots and other adverse outcomes.* Results from these analyses are presented in Table 3. Statistically significant associations with modest effect sizes were found between true cord knots at birth and all of our secondary outcomes except for caesarean section. We were not able to look at the association between true knots at birth and low Apgar scores at 1 minute. No evidence of small study effects was seen for our main outcome of stillbirths in studies of true knots; Egger's test gave a p value of 0.27 (Fig 6).

*Abnormal coiling and intrauterine fetal death.* We were unable to perform meta-analysis to analyse the relationship between abnormal coiling of the umbilical cord and stillbirth

*Abnormal coiling and other adverse outcomes.* Outcome data are shown in Table 4. For all coiling analyses the hypo- or hypercoiled group was compared to the group with normal coiling only. The only exception is a study of hypocoiling by Strong, Finberg & Mattox [46] where the control group was all cords with an umbilical coiling index (UCI) above the 10th centile (so would also have included hypercoiled cords). Analyses were also performed combining all thresholds for hypo- and hypercoiling as in most cases variation was minimal; for UCI at birth the range of thresholds classed as hypocoiling was from <0.6 to <0.17 UCI, with one outlier [47] using <0.26 UCI. For hypercoiling the range was >0.26 to >0.48 UCI. Some studies

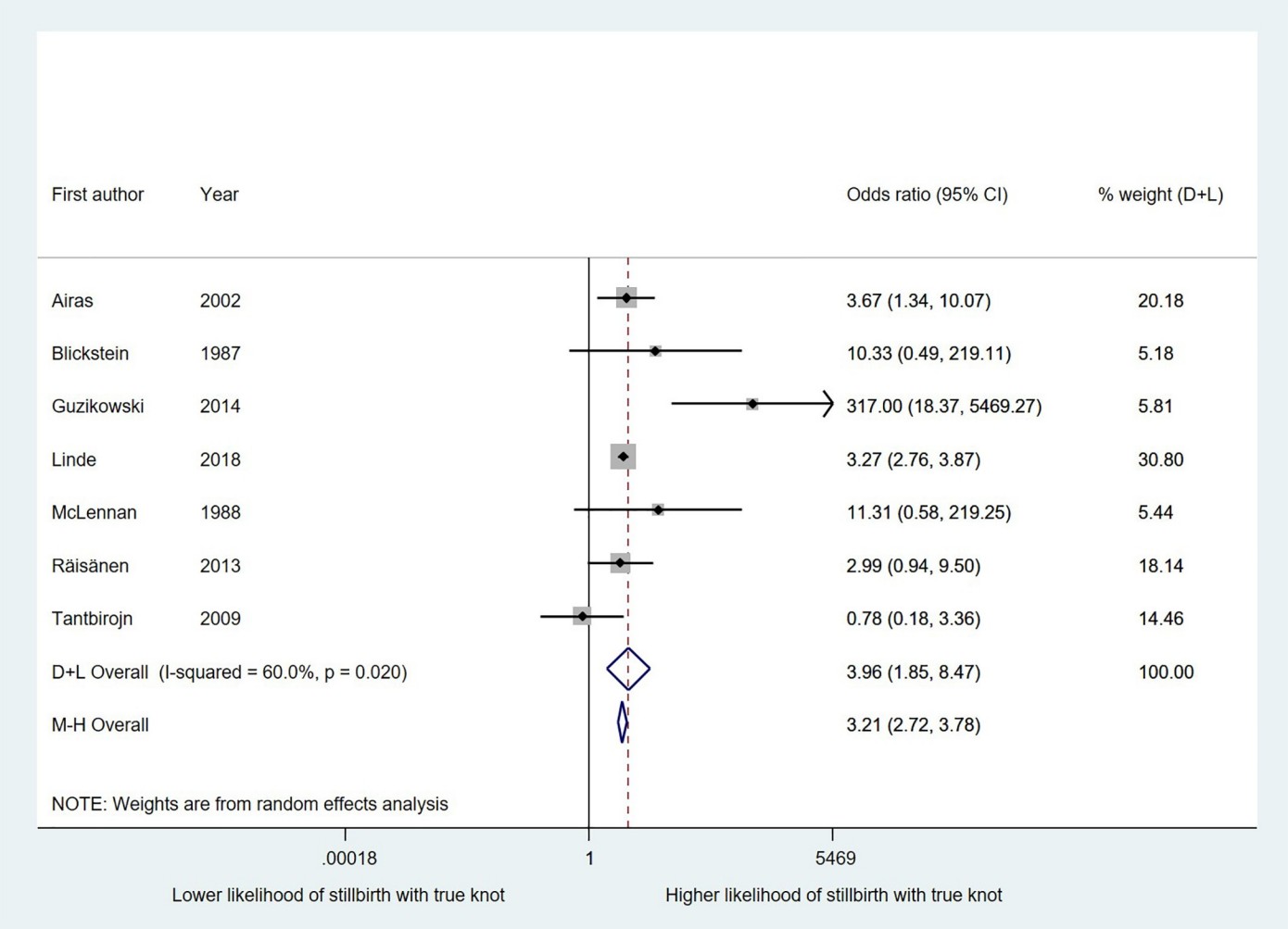

**Fig 5. The association between true knots and the likelihood of stillbirth.**

**Table 3. Analyses of the relationship between true knots and adverse outcomes.**

| | Stillbirth | CS | PTB | 5 min Apgar score <7 | NICU admission | BW<2500g | SGA |
|---|---|---|---|---|---|---|---|
| Odds ratio (95% CI) | **3.96 (1.85, 8.47)** | 1.21 (0.96, 1.51) | **1.15 (1.05, 1.25)** | **1.53 (1.15, 2.03)** | **1.27 (1.06, 1.54)** | **1.31 (1.08, 1.58)** | **1.17 (1.10, 1.24)** |
| $I^2$ | 60.0% (p<0.05) | 66.8% (p<0.005) | 1.6% (p = 0.362) | 27.4% (p = 0.239) | 37.0% (p = 0.190) | 0.0% (p = 0.929) | 0.0% (p = 0.502) |
| Number of pregnancies | 911,814 | 985,919 | 907,024 | 916,669 | 915,721 | 119,987 | 974,325 |
| Notes | All knots detected at birth. 7 studies of 3631 stillbirths. | Mixture of EmCS and all; knots never an indication for CS | Three studies, one 92.27% of weight | Data from five studies | Data from four studies | Data from three studies | Data from four studies |

Shading and bold text indicate statistical significance. **Key:** BW = birth weight, CS = caesarean section, EmCS = emergency caesarean section, NICU = neonatal intensive care unit, PTB = pre-term birth, SGA = small for gestational age.

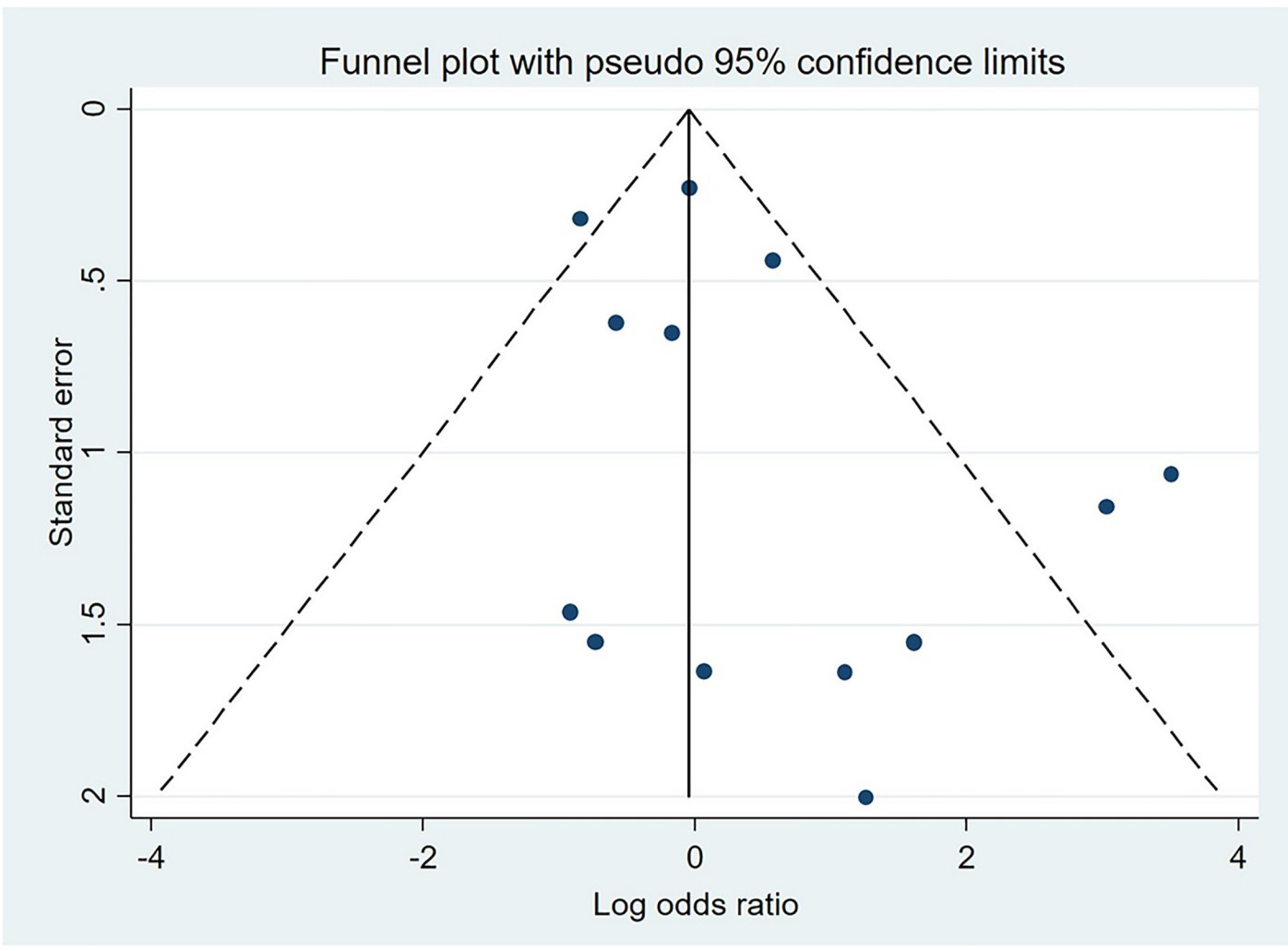

**Fig 6. Funnel plot for studies included in the analysis of the relationship between nuchal cord and stillbirth.**

stated that they used $<10^{th}$ and $<90^{th}$ centile thresholds but did not specify the actual measurements to which these corresponded. Studies that measured UCI antenatally were not analysed with studies that measured coiling at birth; definitions for hypo- and hypercoiling from these studies also tended to differ, potentially due to the gestation at measurement.

Apgar score $<7$ at 1 minute was measured with UCI at birth by three studies [48–50] and antenatal UCI by another [51]. One study defined a low Apgar score at 1 minute as below 4 [52] and another combined all poor Apgar scores [47]. These studies were not included in this analysis. Thresholds for diagnosis of hyper- or hypocoiled cords are displayed on the forest plots and did not appear to lead to any variation in the effect sizes between studies. Sensitivity analyses were also performed for the SGA/FGR analyses based on whether a definition for this outcome was provided by the study, but no effect was seen. For hypercoiling the OR was reduced once studies with unclear definitions were removed but there was still a statistically significant association with SGA.

*Abnormal cord length and adverse outcomes.* We did not perform any analyses of the relationship between abnormal cord length and adverse outcomes due to variation in study definitions, as described earlier.

**Table 4. Analyses of the relationship between abnormal coiling and adverse outcomes.**

|  | PTB <37w | 5 min Apgar <7 | 1 min Apgar <7 | NICU admission | BW <2500g | SGA/FGR (all definitions) | CS |
|---|---|---|---|---|---|---|---|
| Hypocoiling |  |  |  |  |  |  |  |
| Odds ratio (95% CI) | **1.68** (1.18, 2.39) | **4.45** (2.04, 9.71) | **3.18** (1.36, 7.43) | 2.24 (0.83, 6.04) | 2.06 (0.91, 4.70) | 1.65 (0.76, 3.58) | **2.82** (2.13, 3.73) |
| I$^2$ | 0.00% (p = 0.800) | 78.6% (p = 0.00) | 62.5% (p = 0.069) | 74.9% (p<0.01) | 81.2% (p = 0.00) | 71.6% (p = 0.00) | 0.0% (p = 0.533) |
| No. of pregnancies | 2396 | 3982 | 1547 | 2123 | 2578 | 6150 | 4751 |
| Hypercoiling |  |  |  |  |  |  |  |
| Odds ratio (95% CI) | **2.48** (1.52, 4.06) | 1.97 (0.93, 4.18) | 0.80 (0.45, 1.43) | 1.56 (0.85, 2.86) | **3.69** (1.79, 7.64) | **4.31** (1.89, 9.79) | **3.55** (1.74, 7.26) |
| I$^2$ | 43.0% (p = 0.135) | 78.5% (p = 0.00) | 0.00% (p = 0.707) | 32.4% (p = 0.193) | 78.9% (p = 0.00) | 84.3% (p = 0.00) | 82.0% (p = 0.00) |
| No. of pregnancies | 2275 | 3969 | 1550 | 2118 | 2781 | 6767 | 4820 |

**Key:** BW = birth weight, CS = caesarean section, FGR = fetal growth restriction, NICU = neonatal intensive care unit, PTB = pre-term birth, SGA = small for gestational age.

*Cord prolapse and adverse outcomes.* We were unable to perform meta-analysis to investigate the relationship between cord prolapse and any of our outcomes of interest due to a lack of available data.

A summary of normal cord characteristics, UCA incidences, diagnostic accuracy data, and associations between UCA and stillbirth is provided in Table 5.

**Table 5. Summary of results.**

| Normal characteristics | | |
|---|---|---|
| Cord length at birth | | 56.0±11.1cm |
| Cord length at birth (39 weeks) | | 55.6±12.4cm |
| Umbilical coiling index at birth | | 0.24±0.10 coils/cm |
| **Incidence of UCA** | | |
| Nuchal cord at birth | Any | 22% (19, 24%) |
|  | One loop | 16% (13, 19%) |
|  | Two loops | 3% (2, 4%) |
|  | Three loops | 1% (0, 1%) |
|  | Four or five loops | <1% |
|  | Loose loop | 10% (4, 18%) |
|  | Tight loop | 5% (4, 7%) |
| Nuchal cord detected antenatally | Any | 28% (21, 36%) |
| Cord prolapse | | 0.17% |
| True knots | At birth | 1% (0, 1%) |
| **Diagnostic accuracy of ultrasound** | | |
| Nuchal cord | All gestations | Sensitivity 80.5 (66.3, 89.6) |
|  |  | LR+ 6.01 |
|  |  | LR- 0.17 |
|  |  | DOR 26.6 (9.46, 74.7) |
|  | Early labour | Sensitivity range 90.2 to 96.8 |
| **Associations between cord abnormalities and stillbirth** | | |
| Nuchal cord at birth | Any nuchal cord | OR 1.11 (0.62, 1.98) |
|  | Single loop | OR 0.87 (0.56, 1.35) |
|  | Multiple loops | OR 2.36 (0.99, 5.62) |
| Nuchal cord detected antenatally | Any nuchal cord | OR 0.72 (0.17, 3.05) |
| True knots at birth | Any | OR 3.96 (1.85, 8.47) |

## Discussion

Our systematic review was able to combine a large amount of data to determine the normal characteristics of umbilical cord and report the frequency of abnormalities. Some cord abnormalities are common, for example nuchal cord was found in 22% of births (95% CI 19–24), whereas true knots and cord prolapse are less common (1% and 0.1% of births respectively). The definition of some abnormalities e.g. UCI were consistent between studies but others, such as the length of cord were heterogeneous, due to the thresholds applied to define abnormality which often overlapped with the normal ranges (e.g. pooled mean cord length 56.0±11cm, "long cord" defined as >59cm). Estimates of frequency also varied by gestation studied.

We selected stillbirth as our primary outcome, and identified secondary outcomes to reflect diagnosis of fetal compromise that was not sufficiently severe to cause fetal death or represent intervention that prevented it. None of the abnormalities studied showed a significant association with all outcomes and the observed odds ratios were in the range of 1 to 5. This review found the diagnostic accuracy of antenatal or antepartum ultrasound to identify cord abnormalities was modest; the diagnosis of nuchal cord was most accurate when performed in early labour (all studies had sensitivity >90% and specificity >83%). This may be because its incidence is highest at term and there is less time for fetal movements to affect whether the cord is around the fetal neck or not [36, 53]. There are insufficient data to determine whether other abnormalities of the umbilical cord can be reliably detected by antenatal ultrasound.

### Strengths and limitations

This systematic review was strengthened by being conducted according to a pre-specified protocol by an international multidisciplinary review team to maximise the inclusion of relevant data. Up to 270,973 births were included in the meta-analyses giving robust estimates of effect size. However, this review is limited by variation in the definitions used to define both UCA and the associated outcomes, which restricted the number of analyses that can be reliably performed. We were also not able to identify any unpublished data suitable for inclusion, meaning that some of our effect sizes may be overstated due to publication bias.

Our proposed pathway for the differential effects of chronic vs. acute UCA is also limited in that we could mostly report evidence of UCA at birth without knowing how long it had been present, and could not address temporal variation (i.e. whether nuchal cord had been intermittently present). Due to the nature of our included studies we were also unable to distinguish between events that occurred antepartum vs. intrapartum or acute vs. chronic effects, nor could we look at the effects of combinations of UCA, which could affect the likelihood of adverse outcomes, for example shorter cords may lead to tighter nuchal cords and knots when they are present while longer cords may be more prone to entanglements (studies have shown the average cord length to be higher in cases with nuchal cord) [54]. Limb and body entanglements were also not recorded by the majority of studies.

Quality assessment showed that recording of UCA needs to be far more stringent, especially in nuchal cords; number of loops, tightness (which is unlikely to be a true dichotomous variable [55] and also may change during labour so tightness at birth may not reflect tightness antenatally) [56], and type of nuchal cord (A or B, indicating whether the cord is in a locked pattern or can easily be unwound) [57] should all be recorded along with whether other entanglements were present. Classification of stillbirth also requires improvement so that umbilical cord pathology is accurately recorded. Early classification systems such as the Wigglesworth classification did not include umbilical cord complications as a cause of perinatal death. Even when modern classification systems are applied, there is variation in recording of umbilical cord complications resulting in the estimated incidence of cord complications varying from

3.4% to 20%. A recent detailed analysis using the INCODE system suggested 19% of stillbirths are due to cord accident [58]. Variation in reporting of umbilical cord pathology would be reduced by a core outcome set for studies examining the association between UCA and adverse outcome.

## Clinical implications

Our data demonstrate that UCA are associated with adverse perinatal outcomes. The broadest range of associations with stillbirth and associated adverse outcomes were seen for true knots, following by coiling abnormalities then nuchal cord, whereas the strongest effect sizes were for tight nuchal cords. Robust information about the diagnostic accuracy for UCA is only available for nuchal cords, in this case the pooled sensitivity and specificity of antenatal ultra-sound was 80.5% and 86.6% respectively. Nuchal cords were only associated with adverse outcomes when either multiple or tight loops were present. On the basis of this information identifying an isolated nuchal cord antenatally is unlikely to prevent adverse outcome, but may increase intervention. However, combining identification of nuchal cord and abnormal umbilical artery flow increases the likelihood of intrapartum compromise [7, 59]. Further test-accuracy studies are needed, but must be appropriately blinded to prevent intervention altering the outcome.

We hypothesised that umbilical cord abnormalities act via a common pathway of restricting blood flow to the fetus which may be acute or chronic (Fig 1). Thus, we expected to see associations with stillbirth and the secondary outcomes investigated in this meta-analysis. Given the comparatively modest effect size of the relationship between UCA and the outcomes studied here, we conclude that all stillbirths and adverse perinatal outcomes should be thoroughly investigated, even when UCA are present at birth to determine whether a) histopathological changes consistent with UCA are present, including lesions of fetal vascular malperfusion [42], and b) to exclude other possible causes, in order that a robust link may be made between the outcome and antecedent cord complications. In the context of stillbirth, the triple risk model proposes that fetal deaths can result from a combination of fetal stressors, maternal factors, and placental or fetal vulnerability [60]. Applying this model, UCA is a fetal stressor, where stillbirths occur with combinations of risk factors such as reduced placental perfusion, and maternal factors such as maternal obesity, or maternal sleep position. Further studies are also needed to understand the biological mechanisms underpinning UCA and adverse outcomes. For some, such as tight loops of cord or a true knot, this may be from direct occlusion, whereas in hyper- or hypo-coiled cords this may reflect haemodynamic consequences or developmental abnormalities. Larger datasets applying consistent thresholds for abnormalities are required to accurately determine the relationship of UCA to adverse perinatal outcomes.

## Conclusion

This systematic review and meta-analysis has demonstrated links between UCA and several adverse pregnancy outcomes, although not all analyses were adequately powered and some comparisons were restricted by the methodologies of the original studies. Further studies are needed to allow robust clinical recommendations on the management of UCA to be made. These should make use of the information presented about normal cord characteristics to inform thresholds for abnormalities and examine multiple UCA and a range of adverse perinatal outcomes. Ideally, UCA should also be recorded antenatally in blinded studies so that prognostic accuracy can be calculated. Until such data are available, clinicians should be cautious about assigning causality of an adverse outcome based on an isolated observation of UCA.

## Supporting information

**S1 Checklist.**
(DOC)

**S1 Appendix.**
(DOCX)

**S1 Data.**
(XLSX)

**S2 Data. QUADAS-2 for studies of ultrasound accuracy for detection of nuchal cord.**
(DOCX)

## Author Contributions

**Conceptualization:** Dexter J. L. Hayes, Alexander E. P. Heazell.

**Data curation:** Dexter J. L. Hayes, Jane Warland, Mana M. Parast, Robert W. Bendon, Junichi Hasegawa, Julia Banks, Laura Clapham.

**Formal analysis:** Dexter J. L. Hayes.

**Methodology:** Dexter J. L. Hayes, Alexander E. P. Heazell.

**Writing – original draft:** Dexter J. L. Hayes, Alexander E. P. Heazell.

**Writing – review & editing:** Dexter J. L. Hayes, Jane Warland, Mana M. Parast, Robert W. Bendon, Junichi Hasegawa, Alexander E. P. Heazell.

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
