## [Decision Letter · Decision Letter 0]

30 Jun 2020

PONE-D-20-15840

Umbilical cord characteristics and their association with adverse pregnancy outcomes: a systematic review and meta-analysis

PLOS ONE

Dear Dr. Hayes,

Thank you for submitting your manuscript to PLOS ONE. After careful consideration, we feel that it has merit but does not fully meet PLOS ONE’s publication criteria as it currently stands. Therefore, we invite you to submit a revised version of the manuscript that addresses the points raised during the review process.

Please address all reviewer comments including those in the provided attachment.

We look forward to receiving your revised manuscript.

Kind regards,

Kelli K Ryckman

Academic Editor

PLOS ONE

Journal Requirements:

Reviewers' comments:

Reviewer's Responses to Questions

**Comments to the Author**

1. Is the manuscript technically sound, and do the data support the conclusions?

Reviewer #1: No

Reviewer #2: Yes

Reviewer #4: Yes

2. Has the statistical analysis been performed appropriately and rigorously? 

Reviewer #1: No

Reviewer #2: Yes

Reviewer #4: Yes

3. Have the authors made all data underlying the findings in their manuscript fully available?

Reviewer #1: Yes

Reviewer #2: Yes

Reviewer #4: Yes

4. Is the manuscript presented in an intelligible fashion and written in standard English?

Reviewer #1: Yes

Reviewer #2: Yes

Reviewer #4: No

5. Review Comments to the Author

Reviewer #1: 1. The heterogeneity of the studied included (see lines #143-148) (and graded subjectively by the authors as poor (n=7), fair (n=103), and good (n=35), in my assessment negates any statistical analysis in aggregate (essentially, the gist of this submission). I note the authors themselves acknowledge this major drawback (see line # 162 "Heterogeneity was very high").

2. The primary outcome criteria assessed "stillbirth" (see line # ), clearly falls short of the outcome as suggested in the title "adverse perinatal outcome". The authors later in the manuscript address "fetal compromise" (see lines # 351-361). Tools applied in these analyses are incomplete (see next point). The statement "and the observed effects sizes were comparatively modest" is unsubstantiated and lacks any clinical significance.

3. Apgar scores are highly subjective. More precise and scientifically correct perinatal outcome include umbilical artery pH and base excess, both absent in the data assessed in this submission.

4. It is methodologically incorrect to include cord prolapse in the context of "umbilical cord abnormalities" (see lines # 180-181 and later lines # 338-340), (often associated with artificial rupture of the membranes - ie, iatrogenic) in the analysis of nuchal and true knots of the umbilical cord (both spontaneous events).

5. Data pertaining to umbilical cord length are incomplete.

6. Data pertaining to prenatal sonography of nuchal cords and true knots of the umbilical cord, are incomplete.

7. The statement in line # regarding the relative higher sensitivity rates associated with intrapartum prenatal sonographic diagnosis (lines # 220-222) reflect the shorter time from diagnosis to delivery. Worded differently, a nuchal cord throughout gestation is a dynamic condition, often not seen at repeat sonographic assessment (and conversely).

8. I agree with the authors that tightness of the umbilical cord is highly subjective and cannot be addressed (lines # 246 -248).

9. The increased likelihood of stillbirth in cases of true knot of the umbilical cord (lines # 282-285), is well-established.

10. Lines # 363-369): Respectfully, I differ with the authors, in that this "systematic review" is weak (the fact that a pre-specified protocol by an international multidisciplinary review team, is irrelevant). A weak paper cannot be strengthened by these measures. In contrast, I do agree with the authors regarding the notable limitations of this submission - namely the heterogeneity of the included studies reflected in the variations of definitions utilized and associated outcomes.

11. Similarly, as the authors correctly state (lines # 368-369), all data presented may reflect publication bias.

12. Clinical implications stated by the authors (lines # 389-413), are well recognized within the Obstetrical community and do not reflect original previously unpublished data.

13. Line # 84: "Death of a baby" should be simply "stillbirth", or "intrauterine fetal demise".

14. Results: (line # 28): A sentence should not begin with numbers.

15. The final sentence of the Results section of the Abstract (lines # 34-35) is incomplete as written.

16. Abstract: The final comment regarding the need for future studies (lines 39-40), merely reflects the weakness of the secondary outcomes assessed.

17. The first sentence of the Conclusion (line # 37) is incorrect as written. Pregnancy itself "is associated with stillbirth". It appears that the authors intend to state: "True umbilical cord knots are associated with an increased risk of stillbirth". Regarding the remainder of the sentence (lines # 37-38) "the incidence of stillbirth is higher with multiple nuchal loops". Higher than what? Please clarify. It appears the authors mean to state, "in contrast to cases of a single nuchal cord, multiple nuchal loops have an increased risk of stillbirth".

18. The tables are exhaustive, and in my assessment do not contribute significantly to this effort. For example. regarding Lolis et al (1998), why would umbilical cord length increase with parity. I question whether such a "finding" is worthy of quoting?

Reviewer #2: The authors of the manuscript entitled “Umbilical cord characteristics and their association with adverse pregnancy outcomes: a Systematic review and meta-analysis” have produced a fine analysis of literature published in the last 50 years on the impact of umbilical cord abnormalities on stillbirths and other related adverse pregnancy outcomes.

There are a few comments:

1. Robert W Bendon is lacking his institutional address.

2. Introduction: page 4, line 63: duration of UCA: this is most difficult to know/assess (unless cord prolapse). I suggest to delete this element as a feature of variation in published results

3. M&M: page 5, line 87: low birth weight “at term “(<2500g)

4. Table 1: reference number to each paper included should be added

5. Tables 2: clarify at the bottom of the table: NC: nuchal cords

6. Table 3: clarify at the bottom of the Table the meaning of: CS, PTB, NICU, BW, SGA

7. Table 4: clarify at the bottom of the Table the meaning of: PTB, NICU, LBW, SGA/FGR, OR

8. Although Tables 2, 3 and 4 include the significant findings in bold, the readers would benefit from the addition of a Table with the findings related to UCA, which would be better placed at the end of SYNTHESIS OF RESULTS. I.E:

- Loose nuchal loops were more frequent than tight loops, with a summary frequency of 10% (95% CI 4, 18) compared to 5% (95% CI 4, 77).

- Cord prolapse incidence was 0.17%

- True knots at birth incidence was 1%

- etc

Reviewer #4: The manuscript is technically sound, with data supporting the conclusions and rigorous statistical analyses. Data has been made available. However, portions of the manuscript are not well written and need major revision for clarity. Please see attached document for review and further details.

6. PLOS authors have the option to publish the peer review history of their article (what does this mean?). If published, this will include your full peer review and any attached files.

Reviewer #1: No

Reviewer #2: **Yes: **Marta C Cohen

Reviewer #4: No

---

## [Author Response · Author response to Decision Letter 0]

6 Aug 2020

Responses to reviewers have been submitted as a separate document

---

## [Decision Letter · Decision Letter 1]

10 Sep 2020

Umbilical cord characteristics and their association with adverse pregnancy outcomes: a systematic review and meta-analysis

PONE-D-20-15840R1

Dear Dr. Hayes,

We’re pleased to inform you that your manuscript has been judged scientifically suitable for publication and will be formally accepted for publication once it meets all outstanding technical requirements.

Kind regards,

Kelli K Ryckman

Academic Editor

PLOS ONE

Additional Editor Comments (optional):

Reviewers' comments:

Reviewer's Responses to Questions

**Comments to the Author**

1. If the authors have adequately addressed your comments raised in a previous round of review and you feel that this manuscript is now acceptable for publication, you may indicate that here to bypass the “Comments to the Author” section, enter your conflict of interest statement in the “Confidential to Editor” section, and submit your "Accept" recommendation.

Reviewer #2: All comments have been addressed

Reviewer #3: All comments have been addressed

2. Is the manuscript technically sound, and do the data support the conclusions?

Reviewer #2: Yes

Reviewer #3: Yes

3. Has the statistical analysis been performed appropriately and rigorously? 

Reviewer #2: Yes

Reviewer #3: Yes

4. Have the authors made all data underlying the findings in their manuscript fully available?

Reviewer #2: Yes

Reviewer #3: No

5. Is the manuscript presented in an intelligible fashion and written in standard English?

Reviewer #2: Yes

Reviewer #3: Yes

6. Review Comments to the Author

Reviewer #2: Thank you to the authors for a thorough review of their manuscript.

The review provides a very good up-date of literature about umbilical cord abnormalities and stillbirth. This manuscript will be a very useful resource to pathologists and obstetricians.

Reviewer #3: The manuscript can be accepted in its present form as the authors addressed all requets and there are no further changes requested.

7. PLOS authors have the option to publish the peer review history of their article (what does this mean?). If published, this will include your full peer review and any attached files.

Reviewer #2: No

Reviewer #3: **Yes: **Vasilios Pergialiotis

---

## [Editor Report · Acceptance letter]

15 Sep 2020

PONE-D-20-15840R1 

Umbilical cord characteristics and their association with adverse pregnancy outcomes: a systematic review and meta-analysis 

Dear Dr. Hayes:

I'm pleased to inform you that your manuscript has been deemed suitable for publication in PLOS ONE. Congratulations! Your manuscript is now with our production department. 

Kind regards, 

on behalf of

Dr. Kelli K Ryckman 

Academic Editor

PLOS ONE